

# Tidal influences on a future evolution of the Filchner-Ronne Ice Shelf cavity in the Weddell Sea, Antarctica

Rachael D. Mueller[1], Tore Hattermann[2,3], Susan L. Howard[4], Laurence Padman[5],

[1]Earth & Space Research, Bellingham, WA 98225, USA

[2]Akvaplan-niva, Tromsø, 9296, Norway

[3]Alfred Wegener Institute, Helmholtz Centre for Polar and Marine Research, 27570 Bremerhaven, Germany

[4]Earth & Space Research, Seattle, WA 98121, USA

[5]Earth & Space Research, Corvallis, OR 97333, USA

*Correspondence to*: Rachael D. Mueller (mueller@esr.org)

**Abstract.** Recent modeling studies of ocean circulation in the southern Weddell Sea, Antarctica, project an increase over this century of ocean heat into the cavity beneath Filchner-Ronne Ice Shelf (FRIS). This increase in ocean heat would lead to more basal melting and a modification of the FRIS ice draft. The corresponding change in cavity shape will affect advective pathways and the spatial distribution of tidal currents, which play important roles in basal melting under FRIS. These feedbacks between heat flux, basal melting, and tides will affect the evolution of FRIS under the influence of a changing climate. We explore these feedbacks with a three-dimensional ocean model of the southern Weddell Sea that is forced by thermodynamic exchange beneath the ice shelf and tides along the open boundaries. Our results show regionally-dependent feedbacks that, in some areas, substantially modify the melt rates near the grounding lines of buttressed ice streams that flow into FRIS. These feedbacks are introduced by variations in meltwater production as well as the circulation of this meltwater within the FRIS cavity; they are influenced locally





by sensitivity of tidal currents to water column thickness and non-locally by changes in
circulation pathways that transport an integrated history of mixing and meltwater
entrainment along flow paths. Our results highlight the importance of including explicit tidal
forcing in models of future mass loss from FRIS and from the adjacent grounded ice sheet as
individual ice stream grounding zones experience different responses to warming of the
ocean inflow.
**1 Introduction**
The dominant terms in Antarctic mass budget are gains from snowfall and loss by dynamic
flow of ice into the ocean. During the period of 2002-2016, the changes in snowfall were
small (Monaghan et al., 2006) while mass loss from basal melting at the ice-ocean interface
increased (Rignot et al., 2013; Harig and Simons, 2015), largely as a result of changes in the
Amundsen Sea sector (Sutterley et al., 2014). The melting in these floating portions of the
ice sheet, known as ice shelves, translates to a retreat or thinning of the grounded ice due to
a reduction in back-stress ("buttressing") that drives a dynamic acceleration of the grounded
ice (e.g., Scambos et al., 2004; Dupont and Alley, 2005; Rignot et al., 2014). This
acceleration of grounded ice due to ice shelf basal melting has been observed in the Pine
Island, Thwaites, and other glaciers flowing into Pine Island Bay (Mouginot et al., 2014;
Khazendar et al., 2016) and is responsible for the acceleration of Antarctic mass loss
between 2002–2015, as demonstrated by gravity measurements taken by the GRACE
satellite (Harig and Simons, 2015; Groh and Horwath, 2016). Observations (Christianson et
al., 2016; Webber et al., 2017) and models (Joughin et al., 2014) show that the pace of ice
mass loss will depend on changes in ice shelf buttressing, demonstrating a need for
improving predictions of how the extent and thickness of ice shelves will change under
future climate states.



The mass budget for an ice shelf is the sum of inputs from the dynamic flow of ice
across the grounding line and snowfall, and losses from iceberg calving, basal melting,
surface runoff and sublimation, the latter two being insignificant for most Antarctic ice
shelves. Calving and melting each contribute roughly half of the total Antarctic Ice Sheet
mass loss (Rignot et al., 2013; Depoorter et al., 2013), although this ratio varies substantially
between ice shelves in different sectors. The ice shelves that are currently experiencing the
most rapid thinning are in the Amundsen and Bellingshausen seas where relatively warm
Circumpolar Deep Water (CDW) has direct access into the ice shelf cavities (Pritchard et al.,
2012; Rignot et al., 2013). In contrast, the large ice shelves in other sectors that are not
directly influenced by CDW inflows are closer to steady state, suggesting that the transport
of ocean heat under these ice shelves has not changed significantly over the record of our
observations.
We focus here the Filchner-Ronne Ice Shelf (FRIS) in the southern Weddell Sea
(**Fig. 1**). FRIS is a large ice shelf (~430,000 km$^2$) that accounts for 30% of the total area of
Antarctic ice shelves; however, it only contributes 10% of the total ice shelf mass loss. For
comparison, Pine Island Glacier accounts for 0.4% of total area of Antarctic ice shelves but
contributes 7% of total ice shelf mass loss (Rignot et al., 2013). This disproportionately
small melt contribution from FRIS may change in the coming century. Models suggest a
large and persistent increase in ocean temperatures beneath FRIS in response to atmospheric
changes in a warmer, future climate (Hellmer et al., 2012; Timmermann and Hellmer, 2013;
Hellmer et al., 2017). In the modern state, most of the water entering the ocean cavity under
the Filchner Ice Shelf (FIS) and the Ronne Ice Shelf (RIS) is derived from High Salinity
Shelf Water (HSSW) with a temperature close to the surface freezing point of -1.9°C
(Nicholls et al., 2009). Traces of modified Warm Deep Water (WDW) with temperature up
to ~-1.4$^o$C are found in Filchner Trough near the FIS ice front (Darelius et al., 2016), but do
not appear to be a major heat source for melting beneath the ice shelf. In the warming



scenarios of Hellmer et al. (2012), however, almost the entire FRIS cavity is flooded by
WDW with above-zero temperatures by the end of the 21$^{st}$ century, leading to basal melt
rates an order of magnitude higher than present. In this warm state, the associated rapid
thinning of the ice shelf would reduce the buttressing of the large marine-based grounded ice
sheet surrounding FRIS (Ross et al., 2012), significantly accelerating future sea level rise
(Mengel et al., 2016). Furthermore, simulations of Hellmer et al. (2017) show that the
increased meltwater production will sustain the warm inflow, even if atmospheric conditions
were reversed to a colder state, suggesting the existence of an irreversible tipping point once
melting increases past a certain threshold.

This estimate of increased melt, however, assumes that cavity geometry does not

change in a way that alters the access of ocean heat to the FRIS base. Studies of Larsen C
Ice Shelf (Mueller et al., 2012) and Pine Island Glacier ice shelf (Schodlok et al., 2012)
showed that changes to the ice shelf cavity shape can significantly alter the spatial pattern of
basal melt rate, particularly in regions where tidal currents contribute substantially to the
total turbulent kinetic energy near the ice base. Tides were not explicitly included in the
forcing for the Hellmer et al. (2012) study; however, tidal currents play a critical role in
setting the pattern of basal melt rates under FRIS (Makinson et al., 2011), which leads us to
hypothesize that tides would influence changes in meltwater production from a warming
ocean.

We explore this hypothesis using a suite of numerical model simulations that

incorporate variations in tide forcing, initial temperature, and cavity geometry together with
thermohaline interactions at the interface between the ocean and ice shelf. We then use these
models to describe how feedbacks between ice shelf thinning and predicted tidal currents in
the ice shelf cavity influence the evolution of a tide-dominated ice shelf environment under
the condition of increased influx of ocean heat. Lastly, we consider the role of tides on basal
mass loss near the grounding lines of each of the major ice streams supplying ice to FRIS, as



a guide to how individual ice stream grounding zones might respond to the projected
increase in ocean heat flux to the FRIS cavity.
**Methods**
**2.1 Model overview and thermodynamic parameterization**
Our simulations were carried out with a version of the Regional Ocean Modeling System
(ROMS 3.6; Shchepetkin and McWilliams, 2009) that has been modified to include
pressure, friction, and surface fluxes of heat and salt imposed at the base of the ice shelf
(Dinniman et al., 2007, 2011; McPhee et al., 2008; Mueller et al., 2012). ROMS is a
hydrostatic, 3D primitive equation model with a terrain-following (σ-level) coordinate
system and Arakawa-C staggered grid. Our model domain (**Fig. 1**) covers a portion of the
southern Weddell Sea, Antarctica including FRIS. The grid spacing is 5 km with 24 vertical
levels. A full description of model parameter choices and processing options is given in a
supplementary document.
Two model geometries were used in our set of simulations, one representing the modern
state (standard geometry) and the other representing a possible future state (modified
geometry). Model geometry consists of a land mask (including grounded ice sheet), seabed
bathymetry ($h$), and ice draft ($z_{ice}$). These grids are described in **Sect. 2.2**.
Our simulations were initialized with a homogeneous, stationary ocean that has a
potential temperature of either $\theta_{init}$ = -1.9°C ("cold case") or $\theta_{init}$ = -1.4°C ("warm case").
Initial salinity is defined as $S_{init}$ = 34.65 for all cases. The goal of the standard geometry cold
case is to represent present-day temperature and salinity conditions of the primary water
mass entering the ice shelf cavity (Foldvik et al., 2001; Nicholls et al., 2001, 2009), although
our homogeneous representation is a greatly simplified version of realistic conditions. The
warm case represents a moderate ocean warming scenario with an increase of 0.5°C in the



temperature of water being advected into the FRIS cavity. This change is much smaller than
the 2°C temperature increase in the inflowing water by the end of this century predicted by
Hellmer et al. (2012), but was chosen to investigate whether initial feedbacks due to melt-
induced changes in cavity shape from initial warming might be positive or negative. Our
idealized simulations do not include wind forcing, frazil ice, or sea-ice formation.
Circulation develops through buoyancy forcing caused by thermodynamic exchange at the
base of the ice shelves and, for tide-forced cases, by boundary conditions of tidal depth-
integrated velocity and sea surface height. The thermodynamically-driven component of the
circulation was introduced by scalar fluxes at the ice-ocean interface beneath FRIS. These
fluxes are based on the 3-equation parameterization (Hellmer and Olbers, 1989; Holland and
Jenkins, 1999):

$$Q_T^o = \rho_o c_{po} (\alpha_h u_* + m) \Delta T \ [\text{W m}^{-2}], \tag{1}$$

$$Q_S^o = \rho_o (\alpha_s u_* + m) \Delta S \ [\text{kg m}^2 \text{ s}], \text{ and} \tag{2}$$

$$T_b = T_f = 0.0939 - 0.057 S_b + 7.6410 \times 10^{-4} z_{ice} \ [°\text{C}]. \tag{3}$$

In eq. (1), the surface heat flux ($Q_T^o$) is determined by the combined effect of thermal forcing
and turbulent heat exchange. The thermal forcing is represented as $\Delta T = (T_b - T_o)$, where
$T_b$ is the temperature at the ice-ocean interface and $T_o$ is defined as the temperature of the
ocean mixed layer. The value of $T_b$ is assumed to be the freezing point temperature, $T_f$, and
depends on the salinity at the ice-ocean interface, $S_b$, as well as the ice draft, $z_{ice} < 0$ (Foldvik
and Kvinge, 1974; Dinniman et al., 2007). For $T_o$, we follow a common approach of using
the temperature of the surface σ-layer in place of mixed layer values, with the thickness of
the surface σ-layer beneath the ice shelf cavity in our standard grid ranging from 2 to 24 m
and with 72% of points between 5 and 15 m. The turbulent heat exchange at the ice-ocean
interface is represented by a thermal transfer coefficient, $\alpha_h$, scaled by a friction velocity,
$u_*$. This turbulent heat exchange is then adjusted by a meltwater advection term, $m$, that



corrects the scalar fluxes for a computational drift that is introduced as an artifact of
assumptions made in the numerical representation of the ice shelf as a material boundary
(Jenkins et al., 2001). We define $m = -\alpha_s u_* (1 - S_o/S_b)$ where $S_b < 5$ and $m = 0$ elsewhere.
The friction velocity is also calculated from the surface quadratic stress of the upper sigma
level, such that $u_* = C_d^{1/2}|\mathbf{u}|$ with a constant drag coefficient, $C_d = 2.5\times10^{-3}$, and the
magnitude of the surface layer current, $|\mathbf{u}|$. The potential density of seawater, $\rho_o(x,y,z,t)$, is
evaluated for the uppermost layer, with the heat capacity of the ocean, $c_{po}$, given by
$c_{po} = 3985$ J kg$^{-1}$ °C$^{-1}$. $\Delta S$ is the salinity equivalent to $\Delta T$ and is defined as $\Delta S = (S_b - S_o)$,
with $S_b$ solved by quadratic formula from combining Eq. 1, 2, and 3 (without the meltwater
advection term, $m$) and $S_o$ representing the surface σ-layer salinity.

These heat and salt fluxes ($Q_T^o, Q_S^o$) depend on scalar transfer coefficients ($\alpha_h, \alpha_s$) that are

proportional to each other by a "double diffusive" parameter, $R = \alpha_h/\alpha_s$ (McPhee et al.,
2008). Chapter 2 in Mueller (2014) provides a more detailed explanation of the background
and motivation for this parameterization. Here, we used scalar transfer coefficients based on
observations of the Ronne sub-ice-shelf cavity (Jenkins et al., 2010), with $\alpha_h = 1.1\times10^{-2}$ and
$R = 35.5$. The meltwater-equivalent meltrate term is derived by scaling the heat flux, $Q_o^T$, by
latent heat, $L = 3.34\times10^5$ J kg$^{-1}$, and the density of ice, $\rho_i = 918$ kg m$^{-3}$, such that

$$w_b = -Q_o^T L/\rho_i \text{ [m s}^{-1}\text{]}. \tag{4}$$

Melting ice is indicated by $w_b > 0$ and represents the thickness of freshwater added to the
ocean surface, per second. These equations highlight that basal melting is driven by ocean
heat and motion, the latter being influenced by thermohaline circulation and tides.

A set of 12 model simulations was performed and is summarized in **Table 1**. A detailed

description of the different simulations is given in **Sect. 2.3**. Each case involves a



combination of standard or modified geometry, cold or warm ocean, and tidal forcing
switched off or on. For tide-forced simulations, tide heights and barotropic currents were
specified along the domain's open-ocean boundaries (see **Fig. 1**). The tidal boundary
conditions were obtained for the four most energetic tidal constituents, $K_1$, $O_1$, $M_2$, and $S_2$
from the CATS2008b barotropic inverse tide model, an updated version of the circum-
Antarctic model described by Padman et al. (2002). These four constituents account for 94%
of the total tidal kinetic energy for this region, based on CATS2008b estimates. Flather
boundary conditions were used for the barotropic velocity (Flather, 1976) with free surface
conditions following Chapman (1985). Radiation conditions for baroclinic velocities are
applied according to Raymond and Kuo (1984). Tracer equations are radiated across the
open boundaries (Marchesiello et al., 2001) and nudged to initial conditions over a 20-day
time scale.
**2.2 Geometries**
The grid of seabed bathymetry ($h$) over the entire domain (**Fig. 2a**) was derived from the
RTOPO-1 gridded dataset (Timmerman et al., 2010). The ice shelf is represented by a non-
evolving, although freely floating, surface boundary based on an ice draft ($z_{ice}$, **Fig. 2b**) that
was also derived from RTOPO-1. The land mask was adjusted around the ice rises and
rumples in the southern RIS to follow the grounding line provided by Moholdt et al. (2014).
Values of $h$ and $z_{ice}$ in regions of the ice sheet that are grounded in the RTOPO-1 mask but
floating (i.e., ice shelf) in the mask obtained from the Moholdt et al. (2014) data were
computed by linear interpolation and a nearest-neighbor extrapolation to ice shelf points in
the original RTOPO-1 grids.
The ice draft and bathymetry were each smoothed to minimize errors in the baroclinic
pressure gradient term that can occur with terrain-following coordinates as used in ROMS
(Beckmann and Haidvogel, 1993; Haney, 1991). Our Beckman and Haidvogel number, rx0,



is less than 0.045 along both surface and bottom topographies, and our Haney number, rx1,
is less than 10 in both surface and bottom levels except for some areas along the ice shelf
front, where rx1 is larger and reaches a maximum value of 17.
Our maximum values of rx1 are larger than typically recommended for ROMS. To test
whether large values lead to significant circulation from resulting errors in the baroclinic
pressure gradient, we ran unforced models for each of the standard and modified grids. We
initialized these models with horizontally uniform temperature and salinity fields taken from
an extreme stratification profile from the standard warm case. The velocities that develop in
these unforced runs represent possible errors in the full simulations. We calculate grid error
by comparing the currents generated by these uniformly stratified, unforced model runs to
the standard warm tide-forced and modified warm tide-forced cases. From this comparison,
we estimate that the maximum error in our velocity fields is 10% for the standard grid and
5% for the modified grid, but these maxima are isolated to a very limited area north and
northwest of Berkner Island. The relative error over most of the domain is negligible.
In the smoothed standard geometry, the ice draft beneath FRIS ranges from 1537 m at
the deepest part of the grounding line to 11 m at the shallowest point of the ice shelf front.
Small values of ice draft near the ice front are unrealistic, but are a consequence of necessary
smoothing. The region of thinned ice shelf represented by these small values is a narrow
band along the ice front (**Fig. 2b and 2c**). The water column thickness, $wct = h + z_{ice}$, ranges
from 50 m (a specified minimum value, chosen for numerical stability) to 1111 m under
FRIS. In the open ocean, $wct = h$ and has a maximum value of 1914 m.
Using this standard geometry, we conducted simulations for both the cold and warm
cases described in **Sect. 2.1**, with and without tides. The modified geometry was then
created from the output from the two 20-year tide-forced simulations (see **Sect. 2.4**) of the
standard cold and standard warm cases. In creating this grid, we assumed that the RIS and
FIS are both in steady state under present-day conditions (Rignot et al., 2013; Depoorter et




al., 2013; Moholdt et al., 2014) represented by our standard cold case, and that the most accurate simulations of basal melting will be those with tidal forcing included (Makinson et al., 2011). Steady state requires that mass input from lateral ice transport across the grounding line plus snowfall onto the ice shelf is balanced by basal melting and iceberg calving that maintains a constant ice-front position. The difference in local melt between the standard warm and standard cold cases, neglecting any ice dynamical feedbacks, would then be equivalent to the rate of change in thickness of the ice shelf, provided the change in melt rate is not offset by changes in mass inputs to the ice shelf.

We applied the melt-rate imbalance for a period of 50 years to provide a sufficiently large change in $z_{ice}$ to significantly alter the general circulation and tidal currents in the cavity. The resulting modified geometry thins $z_{ice}$ by an average across the ice shelf of 30 m and a mode of .03 m. The ice shelf thickens in the below freezing, mid-shelf regions by a maximum of 14 m and thins in the above freezing, melt regions by a maximum of 453 m (**Fig. 2c and 2d**). The combined area of where the ice shelf thickens is only 0.1% of the total ice shelf area and is characterized by an average increase of 5 m. Given that the modified-case bathymetry is the same as the standard case bathymetry, these changes in $z_{ice}$ cause a change in *wct* of equal magnitude. We chose to only run the modified geometry as a warm case because it is designed to represent the FRIS cavity under warm conditions. Similar to the standard geometry, we ran the modified geometry case with and without tide forcing along the open boundaries.

## 2.3 Model Simulations

Three types of simulations were run: (1) tide-resolving and no thermodynamic exchange; (2) simulations with ice/ocean thermodynamics, with and without tide forcing; and (3) passive dye tracer simulations to explore circulation patterns. These runs are more fully described in the following sections.



### 2.3.1 Tide-resolving simulations with no thermodynamic exchange


We performed two 40-day simulations with 2-hr-averaged output, one with standard
geometry and the other with modified geometry, to predict tidal current speeds. These
simulations ("tides-only cases") did not include thermodynamic interactions at the ice-ocean
boundary, so that the ocean remained unstratified at its initial homogeneous state. Absent
stratification, the resulting currents are barotropic in nature, although some depth-
dependence arises from the friction at the seabed and ice base (see, e.g., Makinson et al.,

2006).

The spatial characteristics of time-averaged tidal currents (**Fig. 3a,b**) were calculated as
the time- and depth-averaged tidal current speed $|\mathbf{u}|_{tide}$, given by:

$$|\mathbf{u}|_{tide} = \langle \sqrt{u_b^2 + v_b^2} \rangle_t \ \ [\text{m s}^{-1}], \qquad (5)$$

where $u_b(x,y,t)$ and $v_b(x,y,t)$ are orthogonal components of modeled, depth-averaged
current and $\langle \rangle_t$ represents temporal averaging over the last $t = 30$ days of the model run,
which characterizes two cycles of the 15-day spring-neap cycles generated by the $M_2$, $S_2$, $K_1$
and $O_1$ tides. The maximum tidal speed at spring tides is, typically, about $2\times|\mathbf{u}|_{tide}$.

### 2.3.2 Base simulations


Six simulations, each 20-30 years long, were conducted with thermodynamic exchanges of
heat and freshwater at the ocean/ice-shelf interface. Output for each of these simulations was
averaged over 30 days. Three of these were run with tidal forcing, two standard geometry
cases, one with $\theta_{init} = $ -1.9°C (cold) and one with $\theta_{init} = $ -1.4°C (warm), and one modified
geometry case with $\theta_{init} = $ -1.4°C (warm). These three simulations all reached steady state
solutions over 20 years, and we refer to them as: standard cold tide-forced, standard warm



tide-forced, and modified warm tide-forced (**Table 1**). We then used the last 30-day
averaged output grids in these tide-forced solutions as initial conditions for three "restart"
simulations without tidal forcing, each of which reached a new steady state over 10 model
years. We refer to the individual runs as: standard cold no-tides, standard warm no-tides, and
modified warm no-tides.
**2.3.3   Passive dye tracer simulations**
Three simulations were run with passive dye tracers to investigate the advection and
diffusion of water from different regions. These simulations were initialized with the steady-
state solutions of the standard cold, standard warm, and modified warm tide-forced cases.
They were run for 2 years each, with 30-day averaged output. Two types of dyes were used.
Passive "meltwater" dyes were continuously added to the model's surface sigma layer at a
rate of $1 \times 10^4 w_b$ in six regions, the grounding zones of five tributary glaciers plus South
Channel (**Fig. 4**). A "bulk" dye was added to the open ocean region shown in **Fig. 4.** The
bulk dye was initialized at a concentration of 100%, but was not replenished after these
simulations began.
**Results**
The main result of our study is that melt-induced changes in cavity shape introduce regional
variations in tide current speeds and advection pathways that result in differing feedbacks to
basal melting. We explain these insights in the following subsections through analyses of:
tidal currents, spatial patterns of $w_b$, ice-shelf-averaged $w_b$ and its sensitivity to model setup,
regionally-averaged $w_b$ and its sensitivity to model setup, and ocean circulation patterns
shown by maps of dye tracer distribution.



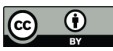
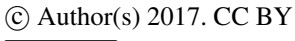

**2.4 Tidal currents in tide-resolving simulations**

The maps of $|\mathbf{u}|_{tide}$ defined by **Eqn. (5)** (**Fig. 3a and 3b**) highlight the spatial variability of tidal currents beneath FRIS. In particular, they show negligible tidal currents in the inlets of the major ice streams that feed into the RIS and FIS, and local maxima along the northeastern RIS front and the South Channel.

The maximum along the northeastern ice shelf frontal zone (ISFZ) of the RIS front is consistent with previous tide models (e.g., Robertson et al., 1998; Makinson and Nicholls, 1999). This region has a relatively small *wct* (**Fig. 2c**), so a larger tidal current here is expected. The melt-induced geometry change in the modified case (**Fig. 3c**) has the overall effect of increasing the water depth in this region and reducing these tidal currents (**Fig. 3d**).

Tidal currents in South Channel are not as strong as those in the northeastern ISFZ, but the melt-induced change in *wct* and $|\mathbf{u}|_{tide}$ is larger than in the ISFZ (**Fig. 3c and 3d**). The melt-induced change in *wct* is also large along the outer grounding line of FIS (**Fig. 3c**). These changes in *wct* and $|\mathbf{u}|_{tide}$ enhance barotropic tidal transport ($|\mathbf{u}|_{tide} \times wct$) along the southern edge of South Channel, the interior grounding line of FIS, and along the western edge of Berkner Island (**Fig. 3e**). These comparisons show that melt-induced ice shelf thinning generally reduces the local tidal currents (**Figs. 3c,d**). However, larger-scale reorganization of the barotropic tidal energy fluxes under FRIS also occurs (see, e.g., Rosier et al. (2014) and Padman et al. (in review)), so that simple scaling of modern tidal currents by the change in *wct* is not possible.

**2.5 Spatial pattern of melt rates ($w_b$) in the base simulations**

The six base simulations described in **Sect. 2.3.2** are analyzed here to quantify the sensitivity of the spatial pattern of basal melt rates ($w_b$) to $\theta_{init}$, tides, and geometry.



### 2.5.1 Base simulation "no-tides" cases


The pattern of $w_b$ in the standard cold no-tides case (**Fig. 5a**) is generally consistent with the
increase in thermal forcing at the ice base $\left( \Delta T = \left( T_f - T_o \right) \right)$ due to the depression of the in
situ freezing point temperature of seawater ($T_f$) as pressure increases. The greatest values of
$w_b$ occur along the deepest grounding lines (see **Fig. 2** for geometry), notably in the Support
Force, Foundation, and Rutford inlets. This pattern of pressure-dependent melt fuels the ice
pump mechanism that drives thermohaline circulation within the cavity and causes
refreezing conditions ($w_b < 0$) by ascending melt water to the mid-ice-shelf regions, as can
be seen under the central RIS and near the RIS ice front. Since our model does not include a
mechanism for frazil-ice formation, these refreezing regions represent where ice would form
by direct accretion to the ice base.
The pattern of $w_b$ for the standard warm no-tides case (**Fig. 5b**) is generally similar to
the standard cold no-tides case (**Fig. 5a**) but with increased melt rates. Changing cavity
shape while imposing the same initial ocean temperature in the modified warm case (**Fig.**
**5c**) only slightly modifies melt rates for the no-tides scenario (cf. **Fig. 5b and 5c**).

### 2.5.2 Base simulation "tide-forced" cases


Adding tide forcing to the standard cold case changes the magnitude and pattern of $w_b$
(compare **Figs. 5a and 5d**). Melt rates increase around the grounding line and, more notably,
in South Channel, where tidal currents are strong (**Fig. 5g**). The increase in $w_b$ in South
Channel between the standard no-tides and tide-forced cases exceeds 2 m a$^{-1}$. Adding tides
also leads to an increase in refreezing in portions of the RIS, including north of Korff Ice
Rise and along the coast of the northeastern RIS. This increase in refreezing with tides can
be explained by the increased production of cold, buoyant meltwater from the deeper parts



of the ice shelf and is qualitatively consistent with the effects of adding tides reported by
Makinson et al. (2011).

The standard warm case follows the standard cold case in that $w_b$ increases around the
deep grounding lines when tides are added (compare **Figs. 5b and 5e**). In the warm case,
however, this increased melt doesn't enhance mid-shelf basal freeze conditions as much as
in the standard cold case. Consequently, the increase of refreezing under the RIS is less
pronounced than for the standard cold tide-forced case (compare **Fig. 5e** with **Fig. 5d**).
There are two possible factors contributing to this result. First, the meltwater product at the
deepest grounding lines in the warm case is warmer than in the cold case and, hence, has a
smaller potential for supercooling when reaching shallower parts of the ice base. Second, the
rising meltwater may warm on its ascent due to admixture of warmer ambient water in the
ice shelf cavity. Both factors are consistent with an increase in thermohaline circulation in
response to warmer temperatures. Similar to the standard warm tide-forced case, the
modified warm tide-forced case (**Fig. 5f**) also exhibits this amplifying effect of tides on
basal melt as the standard warm tide-forced case, although with regional differences when
compared to the standard warm tide-forced case (compare **Figs. 5i and 5h**).

The differences between the tide-forced and no-tides cases for all three model setups
(**Fig. 5g-5i**) show that the principal effect of tides is to increase $w_b$ under FIS and South
Channel, with South Channel exhibiting the largest change between the two.

**2.6 Sensitivity of $w_b$ to tides, $\theta_{init}$ and geometry in the base simulations**

We summarize the effect of $\theta_{init}$ and geometry through values averaged over the ice shelf
area. Net mass change ($M_b$: Gt a$^{-1}$) and averaged values of $w_b$ were calculated for three
regions: (*1*) all of FRIS, (*2*) areas for which melting conditions are predicted ($w_b > 0$), and
(*3*) areas where freezing conditions are predicted ($w_b < 0$). The freeze-only and melt-only





calculations of mass change are influenced by the mean magnitude of $w_b$ as well as the
extent of melting and freezing regions (**Fig. 5**).
Ocean temperature is the dominant control on integrated $M_b$ and ice shelf averaged $w_b$
(**Fig. 6**; **Table 2**). Regardless of tides and geometry, warming the ocean inflow by 0.5°C
increases net mass loss by a factor of ~3-5. The integrated mass gain due to freezing (marine
ice accretion) is insensitive to temperature for the no-tide cases but sensitive to temperature
for the tide-forced cases (**Fig. 6a**), suggesting that accurate predictions of marine ice
accretion requires accurate representation of tidal currents in simulations.
Tides change the effect of ocean heat on $M_b$ and $w_b$. In particular, the addition of tides
to the standard cold case (our approximation to the modern state) increases net freezing by a
factor of four, almost exactly offsetting a factor of ~2 increase in $M_b$ in melt-only regions.
For the warm cases, tides increase total mass loss in melting regions by about 20–40%, with
most of this increase occurring in South Channel and under FIS (see **Fig. 5**). In contrast to
the cold case, the increases in net freezing in the warm cases is small compared with the
increased mass loss, so that the total basal mass loss for FRIS increases significantly when
tides are added to a warm ocean. We attribute this result to production of a warmer plume
near the deep grounding lines as a result of heat in the mixed layer that isn't fully utilized to
fuel melting.
Changing geometry has a much smaller but still significant effect on ice-shelf-
integrated $M_b$ and averaged $w_b$. In the no-tides simulations, the change from standard to
modified geometry causes net mass loss to increase slightly, suggesting a weak positive
feedback, while the tide-forced cases show a slight decrease, or negative feedback (**Fig. 6**).
This change in sign in the mass loss anomaly is driven primarily by the anomalous behavior
of South Channel, and will be discussed in greater detail in **Sect. 3.4 and 4.3**.



### 2.7 Regional sensitivity of $w_b$ in the base simulations

Regional averages of $w_b$ (**Fig. 7**) indicate that basal mass loss near grounding lines of major inflowing ice streams varies by an order of magnitude within a given simulation. Support Force, Foundation and Rutford ice streams show the largest values, exceeding 2 m a$^{-1}$ for the standard cold cases with and without tide forcing. In contrast, modeled melt rates for Möller and Institute ice streams are in the range of 0.28–0.44 m a$^{-1}$ for the standard cold case runs.

For each ice stream inlet, $\theta_{init}$ is the primary control on melt rate near the grounding line, with mean melt rate approximately doubling from the standard cold case to the standard warm case. Tidal forcing is a secondary control that leads to either an increase or decrease in $w_b$ near grounding lines. For Foundation and Rutford ice streams, adding tides reduces area-averaged melt rates, with the relative change being larger for the warm cases with both standard and modified geometry. The largest fractional change in melt rate due to tides occurs near the Rutford Ice Stream grounding line in the modified-geometry warm case, where adding tides reduces mean melt rate by 40% from 7.7 m a$^{-1}$ to 4.6 m a$^{-1}$.

South Channel is an exception to the general result that regional sensitivity of $w_b$ is more strongly affected by changes in $\theta_{init}$ than tidal forcing. In this region, $w_b$ increases by roughly an order of magnitude between the no-tide and tide-forced simulations, whereas the fractional change due to $\theta_{init}$ is much smaller, about a factor of ~2 (**Fig. 7**). South Channel also experiences a large reduction (~30%) in regionally-averaged $w_b$ in the modified warm tide-forced run compared with the standard warm tide-forced run. We attribute this change in $w_b$ to the reduction in tidal currents as geometry is changed (**Fig. 3**).

### 2.8 Ocean circulation within the FRIS cavity in the passive-dye tracer simulations

General patterns of water mass circulation into and under FRIS are demonstrated by output from the two-year simulations with passive-tracer dyes (see **Sect. 2.3.3**). We focus only on



tide-forced simulations because, as discussed in **Sect. 3.4** and described by Makinson et al.
(2011), tidal currents are known to be critical to patterns of basal melting beneath FRIS.

### 2.8.1   Dye tracer circulation in standard cold tide-forced case

The concentration of the open ocean bulk dye tracer in the upper $\sigma$-layer (**Fig 8a**) reveals
that the FRIS cavity has two different sources of heat inflow. The FIS and innermost RIS
cavities are flooded by a southward transport of the open continental shelf waters across the
FIS ice front, whereas the cavity circulation in the northeastern portion of RIS is dominated
by incursions of water from across the RIS front. The latter inflow does not penetrate deep
into the central and southern RIS within two years of simulation, although this is likely to be
an artefact of the omission of the high salinity shelf water that is known to be formed at the
RIS front and is assumed to fuel gravity currents that reach the deep western grounding lines
of the RIS. In our simulations, the water entering through FIS circulates clockwise along the
deep grounding line. After two years, some dye has reached as far west as Carlson inlet;
however, very little of this dye is found under the central RIS north of the ice rises and
rumples. Support Force, Foundation, and South Channel are most directly impacted by open
ocean inflow. This circulation is generally consistent with the progression of ocean warming
in the Hellmer et al. (2012) model.

Meltwater produced near Foundation Ice Stream grounding line (**Fig. 8b**) reveals

similar clockwise circulation. This water reaches the western RIS ice front in about two
years. Meltwater from Foundation inlet flows into all ice stream inlets to the west of
Foundation. The flow of this meltwater through South Channel is limited to the southern
side of the channel.

Meltwater produced in South Channel also reaches all of the western RIS within two

years (**Fig. 8c**), including much of the central region where refreezing occurs. Meltwater
produced in Rutford inlet flows northward to the west of Korff Ice Rise (**Fig. 8d**).





These dye maps demonstrate that water found in the uppermost layer, in contact with
the ice shelf base, in a specific ice stream inlet is a mixture of the incoming high-salinity
ocean water and meltwater that was produced at other inlets further upstream. As an
important consequence, changes in meltwater production in different regions will alter the
meltwater plume characteristics (e.g., temperature) experienced by downstream ice stream
grounding zones. In the following, it will be shown that these interaction of different
grounding zones leads to non-local feedbacks of the melting response to changes in ocean
temperatures and ice shelf geometry, for modeling of which an explicit inclusion of tidal
currents is one of the key ingredients.
**2.8.2   FRIS cavity dye tracer distribution for tide-forced cases**
Comparisons of dye concentration maps after two years of integration for the three tide-
forced simulations (**Fig. 9**) show differences that can be attributed both to $\theta_{init}$ (comparing
standard cold and standard warm cases) and to geometry (comparing standard warm and
modified warm cases).
The stronger cavity circulation introduced by the amplification of net basal melting for
the warmer ocean, $\theta_{init}$ = -1.4°C, increases inflow through the FIS and into the RIS cavity
(upper row of **Fig. 9**). Open-ocean water is present under most of RIS after two years in the
standard warm case. The dye concentration of open-ocean water under the northern portion
of RIS decreases as $\theta_{init}$ increases, indicating that the stronger northward flow of meltwater
in the warm case reduces the contribution of the direct open-ocean inflow to the northern
RIS. This influence of $\theta_{init}$ on strengthening the sub-ice-shelf cavity circulation decreases
slightly in the modified warm case, which shows less open water dye penetrating into the
innermost RIS than the standard warm case solution (compare the two upper right subplots
of **Fig. 9**).





Comparisons of meltwater dyes from ice stream inlets and South Channel (lower six
rows in **Fig. 9**) show, in all cases, more rapid ventilation of downstream regions when $\theta_{init}$ is
warmer. In these simulations, meltwater dyes are injected continuously at a rate that is
scaled to the basal melt rate (**Sect. 2.3.3**). Relative dye concentrations at specific locations
can, therefore, be interpreted as the relative values of meltwater from different sources with
total meltwater plume concentration being an integration of the contributions from all
upstream source. Changes in the different runs reflect the response of the cavity circulation
and changes in meltwater production rate in the respective grounding zones. Meltwater from
South Channel dominates the central RIS, although melting in Foundation and Rutford inlets
provides a substantial freshwater flux to the western RIS.
The changes in upper-ocean circulation caused by changes in geometry, seen by
comparing the two warm cases in the last two columns of **Fig. 9**, are less obvious than the
effect of changing temperature. Nevertheless, changing geometry has a significant regional
effect. Foundation inlet meltwater spreads out more in the modified warm case than in the
standard warm case as a result of increased dye transport through the channel between
Henry Ice Rise and Berkner Island. South Channel meltwater concentrations are reduced in
the modified warm case, which is consistent with the reduced melt rates in the region
(**Fig. 5**). Similar to South Channel, Rutford inlet meltwater in the surface layer is also
reduced for the modified geometry case.
**2.8.3    Regional meltwater dye comparison, for tide-forced cases**
Regional meltwater dye production and advection is evaluated from the surface levels of
Foundation, Möller, and South Channel inlet regions (as in **Fig. 4**). **Fig. 10** shows the
integrated values over these regions for the standard cold, standard warm and modified
warm tide-forced cases. As described in **Sect 2.3.3**, meltwater dye from a particular region is
a scaled quantity of $w_b$ that reflects the magnitude of meltwater produced in that region. The



resulting passive dye tracer is then transported through the domain through a combination of
advection and mixing and acts as a proxy for the meltwater plume. In this section, we use the
quantity of these meltwater tracers to demonstrate how meltwater circulation is affected by
changes in $\theta_{init}$ and geometry.

489   Foundation inlet shows an expected increase in integrated meltwater dye with the 0.5°C

increase in $\theta_{init}$ between the standard cold and standard warm cases (**Fig. 10a**). This increase
in integrated meltwater dye is not sustained with the change in cavity geometry. Instead, the
net amount of dye is reduced in the modified warm case such that the value of integrated dye
in the surface level more closely matches that of the standard cold case. This reduction in
integrated meltwater dye between the standard warm and modified warm case carries
forward into the Möller region, where the reduction of Foundation dye between the two
cases is even greater than in Foundation (compare **Fig. 10a** and **10b**). At the same time, the
reduction in Foundation dye in the Möller region is somewhat compensated by the Möller
meltwater dye, which is consistent between the two warm cases (**Fig. 10b**). Overall, the
Möller region appears to be less affected by the change in geometry than the Foundation
region. Within the South Channel, the influence of geometry on the quantity of meltwater
dye is compensated by changes in circulation that allow for more Foundation dye in the
surface level of South Channel in the modified warm case than the standard warm case (**Fig.**
**10c**). This increase in surface level Foundation dye in South Channel is caused by changes
in circulation that distribute the dye more evenly across South Channel in the modified
warm case than in the standard warm case (**Fig. 9**).

506   These results highlight that the regional sensitivities of meltwater dye to $\theta_{init}$ and

geometry may influence but not necessarily determine the quantity and quality of meltwater
in downstream regions. This result is important because it reveals the degree to which $\theta_{init}$
and cavity shape precondition the quantity and origin of meltwater in any given region. For
example, the FRIS-integrated surface dye quantity (**Fig. 10d**) for Foundation and Support





Force is equivalent between the standard warm and modified warm cases, even though there
are strong regional variations in these cases (**Fig. 10 a-c**). In addition, the FRIS-integrated
values of meltwater dyes from the ice front regions (RIS west, RIS east, and FIS) are similar
among all cases while they differ among regions, showing greater amounts of dye in
Foundation, Möller, and South Channel regions for the warm cases than in the cold case.
These regional and integrated changes demonstrate that the FRIS meltwater product is a
result of regional feedbacks that are affected by a combination of production, mixing, and
advection.
**Discussion**
**2.9 Comparison of modeled, ice shelf averaged basal melt estimates with observations**
The melt rate averaged over the area of an ice shelf is a common metric for evaluating ice
shelf mass balance (e.g., Rignot et al., 2013; Depoorter et al., 2013). Our estimate of melt
rate averaged over FRIS for the standard case is 0.14 m a$^{-1}$, equivalent to ~48 Gt a$^{-1}$ of net
mass loss (**Fig. 6** and **Table 2**). The range of values reported by other studies extends from
the lower bound in Depoorter et al. (2013) of 0.03 m a$^{-1}$ to 0.55 m a$^{-1}$ for the first
oceanographically-derived estimates reported by Jenkins (1991) and Jacobs et al. (1992); see
**Fig. 11** and **Table 3**. Compared with the three most recent satellite-constrained estimates,
our value is near the central estimate of 0.12 m a$^{-1}$ of Depoorter et al. (2013), and near the
lower limit of the ranges reported by Rignot et al. (2013) and Moholdt et al. (2014).

The range in estimates of $w_b$ is a result of variations in observation type and model

choices. Estimating $w_b$ from observations typically requires averaging other ice shelf mass
budget terms, derived from satellite observations and atmospheric models, over several
years. Estimates from models are affected by model setup. Our idealized model lacks the
seasonal warming of the upper ocean near the ice front that leads to significant summer



melting and rapid basal melting of the ice shelf frontal zone (e.g., Makinson and Nicholls,
1999; Joughin and Padman, 2003; Moholdt et al., 2014). The reduced melt in the frontal
zones in our model helps to explain why our ice-shelf-integrated mass loss by melting is
smaller than in most observations. The lack of an annual cycle of forcing in our model might
also affect our representation of inflows across the ice front. For example, high-salinity shelf
water (HSSW) inflow across the western Ronne ice front is believed to be modulated not
only by the annual cycle of HSSW production in the Ronne Depression but also by seasonal
changes in the vorticity constraint at the ice front, associated with changing stratification
(Nicholls et al., 2009). Neither of these seasonally-varying processes is included in our
simulations.
**2.10 Sensitivity of $w_b$ to $\Delta T$ and surface currents**
In this section, we explore the regional variations of thermal forcing ($\Delta T$) and turbulent
exchange on $w_b$ using the values from the 30-day averaged output of each simulation to
calculate $\Delta T$ and $|\mathbf{u}|$. Note that $w_b$ in the 30-day averaged model output is based on the
average of instantaneous heat fluxes and, therefore, includes the model's knowledge of
covariances between $\Delta T$ and $|\mathbf{u}|$ on much shorter time scales than $|\mathbf{u}|$, which is based on 30-
day averaged u- and v-velocities. We use a linear combination of non-tidal and tidal
currents, $\mathsf{U}$, given by
$\mathsf{U} = |\mathbf{u}|_{tide} + |\mathbf{u}| \ \ [\text{m s}^{-1}]$                              (6)
to represent the local forcing for turbulent exchange, where $|\mathbf{u}|_{tide}$ is from **Eqn. (5)** and $|\mathbf{u}|$ is
calculated from the 30-day averaged output values of u- and v-velocities. For the no-tides
cases, $|\mathbf{u}|_{tide}$ is zero and $\mathsf{U}_{\text{no tides}} = |\mathbf{u}|$. We include $|\mathbf{u}|_{tide}$ in the tide-forced cases to more
closely approximate the non-time-averaged relationship described by **Eqn. (1),** because the
30-day average removes the tidal signal in $\Delta T$ and $|\mathbf{u}|$ in the tide-forced cases.





Comparisons of the six base simulations show that $w_b$ generally follows the expected
functional dependence on $\Delta T$ and $\mathsf{U}$ (**Fig. 12a**): in all six cases, values of $w_b$ increase with
stronger currents and more thermal forcing, with values roughly falling along lines of
constant $\Delta T \cdot \mathsf{U}$. However, regional differences can be seen in the bivariate relationships
between $w_b$ and either $\Delta T$ or $\mathsf{U}$ (**Fig. 12b and 12c**). Most ice-stream inlet averages show a
similar increase in $w_b$ with respect to $\Delta T$ (**Fig. 12b**), suggesting that reasonable estimates of
melt rate in the ice-stream inlets could be obtained from variability of $\Delta T$ and a constant,
assumed low, value of $\mathsf{U}$. South Channel and, to some degree, Institute diverge from this
relationship, demonstrating a larger variability in $w_b$ in relation to $\Delta T$ than is seen in other
inlets (**Fig. 12b**). This larger variability in $w_b$ in South Channel arises because changes in
modeled melt in this area are controlled primarily by changes in $\mathsf{U}$ (**Fig. 12b**).
Comparisons of the ratios for $\Delta T$, $\mathsf{U}$, and $w_b$ at each site between simulations without
and with tides (**Fig. 12d-f**) show how each region responds to the combined effects of tide-
induced changes in ocean conditions. With the exception of South Channel, adding tides
always cools (decreases $\Delta T$) the upper layer of ocean water adjacent to the ice base (**Fig.**
**12d**). On average, the largest reductions occur for the RIS ice stream inlets. We attribute this
result to cooling of water entering the RIS inlets by inclusion of meltwater from upstream
freshwater sources, with RIS inlets being influenced by rapid melting in Support Force and
Foundation inlets, and in South Channel (**Figs. 8 and 9**).
The differences between the tide-forced and no-tide cases show up more strongly in the
regionally-averaged comparison of $\mathsf{U}$ (**Eqn. (6), Fig. 12e**). In all regions, the effect of adding
tides is greater for the cold standard cases than for warm standard cases. Since the value of
$|\mathbf{u}|_{\mathrm{tide}}$ in **Eqn. (5)** is the same for the standard geometry runs, this difference represents the
increase in the thermohaline-driven $|\mathbf{u}|$ from the cold to warm cases.
The largest differences in $|\mathbf{u}|$ amongst all three model runs are in Möller, South Channel,
and Institute inlets. For the warm cases, modifying the geometry increases the ratio of $\mathsf{U}_{\mathrm{tide}\text{-}}$



$_{forced}$ / $U_{\text{no tides}}$ for these three regions even though tidal currents decrease (**Fig. 3**) as *wct*
increases. This response implies that $U_{\text{no tides}}$ also declines in the modified geometry case. A
decline in $U_{\text{no tides}}$ in the modified geometry is consistent with a reduction in $z_{ice}$ in the inlet
regions, which would reduce the thermal forcing and, hence, reducing the ice pump
circulation. If true, this feedback is an artifact of our model geometry, which excludes the
possibility of deeper ice that could be exposed when the grounding line migrates due to the
imposed thinning. Corollary evidence for this reduction in ice pump circulation is seen in the
top row of **Fig. 9**.
The role of South Channel melt on cooling downstream ice stream inlets, its sensitivity
to tides, and tidal sensitivity to changing $z_{ice}$ suggest that reliable predictions of change in
modeled $w_b$ in the southern RIS ice streams for future climate scenarios depends on the
correct representation of changes to South Channel geometry.
**2.11 Role of advection through South Channel**
As the maps of surface-layer dye tracers (**Fig. 8 and 9**) show, most water entering the FRIS
cavity in our simulations flows southward under the FIS front and then circulates clockwise
around the FIS and RIS grounding lines. A water parcel takes about two years to travel from
the FIS front to the southwestern RIS region of Rutford inlet. During that time, each water
parcel is subjected to mixing with meltwater, so that the properties of water entering each
inlet depend on the processes along the entire inflow path. This circulation is driven only by
thermohaline circulation modified by tides; recall that our model excludes the influences of
wind-driven circulation and sea-ice formation.
Distribution of dyes also varies in the vertical, shown in **Fig. 13** for a transect taken
across South Channel from the western tip of Henry Ice Rise. The standard geometry
simulations show a core of open water dye along the bottom and northeastern slope of the
trough. Support Force dye is concentrated near the ice base, toward the southwestern end of

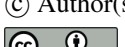



the transect. Foundation dye appears in both the surface and deep model layers, concentrated
on the southern side of South Channel (see, also, **Fig. 8b**). As expected, South Channel dye
has the highest concentration in the surface waters of this transect.

The spatial pattern in dye distribution is fairly consistent between the warm and cold

standard geometry cases, although much more open water dye from north of the ice front is
present in the warm case. This quantitative difference is consistent with the overall
understanding that warmer $\theta_{init}$ drives a stronger thermohaline circulation that enhances
cavity circulation and leads to a shorter residence time (**Fig. 9**).

More qualitative differences between simulations arise from the change in cavity shape.

Except for South Channel dye, the modified geometry shows more laterally uniform dye
concentrations across the channel. Dye distribution remains vertically stratified in all three
cases with the depth of the upper layer being similar in the standard warm and modified
warm cases. However, even though the averaged $w_b$ in South Channel is similar between the
standard cold and modified warm cases (**Fig. 7**), the South Channel meltwater product does
not mix down as far in the modified warm case as it does in the standard cold case (**Fig. 13**),
which we attribute to the much weaker tidal currents in this region (**Fig. 3**) for modified
geometry.

Transects for dye tracers are only provided for the tide-forced cases. However, a

comparison of temperature transects for the no-tides and tide-forced cases (**Fig. 13**) show
that the thermocline is deeper in the standard geometry tide-forced cases, with the
thermocline most affected in the standard cold case. As shown in **Fig. 12d**, tides increase the
thermal forcing in South Channel in the standard geometry by a factor of ~3 while having a
negligible effect on $\Delta T$ in the modified geometry. These results suggest that the lowered
thermocline in this region is caused by tide-induced mixing rather than advection, and in
turn directly responds to the reduced tidal currents in the modified warm case.



### 2.12 Implications of regional melt on ice sheet mass balance

Walker et al. (2008) showed that the spatial distribution of ice shelf melt rates was critical to the behavior of the buttressed grounded-ice streams; for the same integrated mass loss from an ice shelf, grounded-ice loss was significantly faster when the melting was concentrated near the grounding line. Gagliardini et al. (2010) confirmed this analysis, and also noted that a grounded-ice stream could thicken, and its grounding line could advance even when net melting increased, if the melt rate decreased near the grounding line. In the context of our study the implication is that, even when the change in the ice shelf, area-averaged melt rate is not large, substantial variability in melt rates near ice-stream grounding lines could still have a large impact on loss of grounded ice.

In addition to being affected by the spatial distribution of $w_b$, dynamic mass loss of grounded ice is also affected by bedrock slope and ice sheet topography. These factors introduce additional spatial heterogeneity in the influence of basal melting on overall mass loss from the grounded ice streams flowing into FRIS. Wright et al. (2014) used the BISICLES ice sheet model to test the sensitivity of the grounded ice sheet to changes in FRIS mass loss at the grounding line. They found that Institute and Möller ice streams are the most sensitive to changes in basal mass balance that might be caused by a warming ocean inflow. This result was confirmed by Martin et al. (2015) using the Parallel Ice Sheet Model (PISM). These two ice streams rest on top of steep reverse bed slopes with low basal roughness, conditions which have been shown to contribute to grounding line instability and retreat (Schoof, 2007). Furthermore, these ice streams are also sensitive to changes in the buttressing effect from ice shelf mass loss around Henry and Korff ice rises and an increase in basal sliding over these ice rises. Our results indicate that tides currently exert a strong influence on basal mass balance in the area around Henry and Korff ice rises (**Fig. 5**), by increasing melting in South Channel and increasing marine ice accretion north of the ice rises and Doake Ice Rumples.





Möller and Institute are among the lowest meltwater producing regions (**Fig. 7**) and
receive the largest fraction of meltwater product from Foundation inlet basal melt (**Fig. 10**).
The relative quantities of these meltwater products are sensitive to changes in advection and
mixing imposed by changes in $\theta_{init}$ and geometry (**Sec. 3.5**). As shown in **Fig. 7**, the 0.5°C
increase in $\theta_{init}$ increases the Möller grounding-line region $w_b$ to 1.03 m a$^{-1}$ (an increase of
134%) and Institute grounding-line region $w_b$ to 1.30 m a$^{-1}$ (an increase of 100%), for the
tide-forced cases. The grounding lines in these regions appear to be very sensitive to changes
in $\theta_{init}$ and less sensitive to changes in the cavity ocean circulation imposed by a change in
model geometry. Even with Möller and Institute's sensitivity to $\theta_{init}$, however, these inlets
are buffered from variations in open ocean heat due to the combined influence of circulation
pathways and inflowing meltwater derivatives (**Fig. 10**).
Of the nine grounding-line regions explored in this study, Foundation inlet has the
highest averaged melt rate of 2.76 m a$^{-1}$ for the standard cold case (**Fig. 7**), a rate which
more than doubles to 6.01 m a$^{-1}$ when $\theta_{init}$ increases from -1.9°C to -1.4°C. However,
grounded-ice mass flux from Foundation Ice Stream is less sensitive to changes in basal
melting than Möller and Institute (Wright et al., 2014). According to the results presented in
Wright et al. (2014), even the higher melt rate with the warmed ocean in our study is
insufficient to drive grounding line retreat and significant acceleration of grounded-ice loss
at Foundation. Therefore, it is possible that the dominant effect on the grounded-ice mass
budget of large $w_b$ at Foundation is through the effect of Foundation inlet meltwater on
downstream inlets, particularly Möller. As shown in the dye results presented in **Sect. 3.5**,
Möller is somewhat isolated from FIS inflow but flooded by Foundation meltwater.





**Conclusion**
The idealized modeling results presented here on the basal melting of FRIS, combined with
ice-sheet model results reported by Wright et al. (2014), indicate that the response of the
Antarctic Ice Sheet in the Weddell Sea sector to large-scale ocean warming in the Southern
Ocean depends on several regional and local processes that combine to determine ocean
state in individual ice-stream inlets. These processes include the tidal contribution to ocean
mixing, advection of meltwater products into downstream inlets, and feedbacks between
advection, tides and melting as ice shelf draft evolves.
In general, tides increase the area-integrated mass loss from the entire FRIS, consistent
with the findings of Makinson et al. (2011); however, for our cold case ocean representing
the modern state, increased basal melting with tides is completely offset by a factor-of-four
increase in basal accretion (marine ice formation) in the central Ronne Ice Shelf.
As proposed by Hellmer et al. (2012), warming of water entering the cavity under FRIS,
primarily as an inflow under the FIS front, leads to a large increase in ice-shelf-integrated
mass loss. In our simulations, warming of 0.5°C increased total FRIS mass loss by a factor
of ~3.6 for the no-tides simulations (cf. Hellmer et al., 2012) and by a factor of ~5.1 when
tidal forcing was included.
The large-scale, sub-ice-shelf circulation in our idealized model is dominated by a
southward inflow of open-ocean water across the Filchner Ice Shelf front, and clockwise
circulation of this water along the southern grounding line. A water parcel takes about two
years to travel from the Filchner ice front to the southwestern Ronne Ice Shelf. The known,
seasonally varying inflow through the Ronne Depression is not represented in our model,
which lacks the forcing required to drive a seasonal cycle of high salinity shelf water
production and stratification along the Ronne ice front.





At the regional scale, complex feedbacks occur between local processes such as tide-
induced mixing and advection, so that the temperature of a water parcel represents the
upstream integrated history of mixing between the inflowing source water and basal
meltwater. The temperature of the upper ocean layer adjacent to the ice shelf base is cooler
when tide forcing is included, especially in the southern Ronne ice stream inlets (Rutford,
Carlson and Evans). We attribute this cooling to incorporation of meltwater from upstream
sources, notably Foundation inlet and South Channel.
These results show regionally variable responses to changes in tides, $\theta_{init}$, and cavity
geometry that can be summarized as follows.
(1)     Meltwater plumes from basal melting introduce non-local feedbacks, within the
same cavity, in response to variations of inflowing ocean heat and melt-induced
changes in ice draft.

(2)     Tides increase $w_b$ under the FIS, with largest effect within South Channel.
(3)     In some regions (e.g. South Channel), tides influence $w_b$ directly by changing
the friction velocity; in other regions (e.g., Rutford), tides influence meltwater
production through changes in $\theta$ by mixing along the upstream flow path. Tides
affect how $w_b$ changes in response to $\theta_{init}$ and cavity geometry by these direct
and indirect influences.

(4)     The greatest fraction of meltwater in the Möller and Institute inlets are
contributed by Foundation inlet.

The described regional meltwater distribution and $w_b$ is sensitive to the accuracy of our
grids of seabed depth and *wct*, which are based on few passive seismic measurements in
regions of strong model sensitivity (**Fig. 3f**). Distributions are also affected by the model
configuration, including neglect of atmospheric and sea-ice forcing, the choice of mixing
schemes and the thermodynamic exchange coefficients for the ice-ocean boundary layer



parameterization. However, our analysis shows that the interplay of tides, far-field thermal
forcing and the oceanic response to ice shelf geometry changes leads to complex and
sometimes non-local interactions that alter the overall basal mass balance that effects
melting near the grounding lines, thereby controlling the dynamical response of adjacent
grounded ice streams.

A significant source of uncertainty in the future mass loss through the ice streams

draining the West Antarctic Ice Sheet into the Ronne Ice Shelf is in how the ice draft in
South Channel will evolve if the heat flux into the cavity under FRIS increases. Under
modern conditions and with the seabed and ice draft represented by the RTOPO-1 database,
tides are a critical contributor to basal melting in the region. A warmer ocean will increase
mass loss by basal melting that will lead to ice shelf thinning unless it is offset by increased
inputs from ice advection and snowfall. However, this thinning then causes regional
feedbacks that include a reduction in basal melting in South Channel, as tidal currents
weaken, and a change in circulation pathways with consequences for heat and meltwater
transport.

We conclude that it is not possible to predict the true effect of oceanic warming on ice

thinning near individual ice stream grounding lines without a better understanding of the
feedbacks introduced by tidal forcing and circulation as a result of changes in $wct$. That is, as
coupled ocean/ice-sheet models become a standard tool for projecting ice sheet response to
changing climate, tides must be either explicitly modeled, or represented by a
parameterization that itself can evolve with time at a rate set by the evolution of the cavity.
Furthermore, potential bottlenecks in sub-ice-shelf circulation of ocean heat must be
identified through improved surveys of seabed bathymetry which, when combined with the
better-known ice shelf draft, determines both the tidal current speeds and the mean ocean
circulation towards downstream sites including ice-stream inlets. While FRIS is presently in



approximate steady state, the potential for future ocean warming, increased $w_b$, and a
corresponding mass loss causing a ~1 m sea-level rise supports the need to improved
measurements of the seabed bathymetry in the ice stream inlets and under South Channel.
**Author contribution**
R.D.M. led the study. The simulations were designed by R.D.M. and L.P., implemented by
R.D.M. and S.L.H., and analyzed by R.D.M., L.P. and T.H. The paper was written
by R.D.M., L.P. and T.H.
**Acknowledgements**
We thank Mike Dinniman (Old Dominion University) for his invaluable help in developing
ROMS for use in simulating ice shelves and Dr. Scott Springer for his help in creating the
model grid. This study was funded by: NASA grants NNX10AG19G, NNX13AP60G;
NASA Earth and Space Science Fellowship, 07-Earth07F-0095; and The Research Council
of Norway, program FRINATEK, project WARM #231549/F20. This is ESR publication
number XXX.

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





**Table 1** An overview of the eight model runs presented in this paper and the case name that is used to reference them. The three runs that were used as the spin-up solutions to initialize other runs are marked and referenced as b1, b2, and b3. The four runs that include passive dye tracers are shaded in grey. All run intervals provide steady state solutions as determined by the transient solutions of shelf-averaged basal melt.

| # | Case Name | Run length | Averaged output period | Cavity geometry | $\theta_{init}$ (°C) | min $z_{ice}$ (m) | max $z_{ice}$ (m) | min $wct$ (m) | max $wct$ (m) |
|---|---|---|---|---|---|---|---|---|---|
| 1 | standard cold tide-forced | 20-years | 30-day | present-day | -1.9 | -11 | -1537 | 50 | 1210 |
| 2 | standard cold no-tides | 10-years | 30-day | present-day | -1.9 | " | " | " | " |
| 3 | standard warm tide-forced | 20-years | 30-day | present-day | -1.4 | " | " | " | " |
| 4 | standard warm no-tides | 10-years | 30-day | present-day | -1.4 | " | " | " | " |
| 5 | standard tides-only | 30-days | 2-hour | present-day | NA | " | " | " | " |
| 6 | standard cold dye (restart #1) | 2-year | 30-day | present-day | -1.9 | " | " | " | " |
| 8 | standard warm dye (restart #3) | 2-year | 30-day | present-day | -1.4 | " | " | " | " |
| 9 | modified warm tide-forced | 20-year | 30-day | melt-adjusted | -1.4 | -25 | -1442 | 52 | 1180 |
| 10 | modified warm no-tides | 10-year | 30-day | melt-adjusted | -1.4 | " | " | " | " |
| 11 | modified tides-only | 30-days | 2-hour | melt-adjusted | NA | " | " | " | " |
| 12 | modified warm dye (restart #9) | 2-year | 30-day | melt-adjusted | -1.4 | " | " | " | " |





**Table 2** Values for integrated mass transport ($M_b$, Gt a$^{-1}$) and FRIS-averaged basal melt rate ($w_b$, m a$^{-1}$) for the six runs shown in Fig. 6: **Standard cold no-tides** (S -1.9 ⊘), **Standard cold tides-forced** (S -1.9), **Standard warm no-tides** (S -1.4 ⊘), **Standard warm tides-forced** (S -1.4), **Modified warm no-tides** (M -1.4 ⊘), **Modified warm tides-forced** (M -1.4).

|  |  | S -1.9 ⊘ | S -1.9 | S -1.4 ⊘ | S -1.4 | M -1.4 ⊘ | M -1.4 |
|---|---|---|---|---|---|---|---|
| $M_b$ (Gt a$^{-1}$) | Net | 48 | 47 | 171 | 239 | 188 | 221 |
|  | Melt | 62 | 104 | 182 | 262 | 200 | 246 |
|  | Freeze | -14 | -57 | -11 | -23 | -12 | -25 |
| $w_b$ (ma$^{-1}$) | Net | 0.14 | 0.14 | 0.49 | 0.69 | 0.54 | 0.63 |
|  | Melt | 0.27 | 0.46 | 0.66 | 0.95 | 0.69 | 0.86 |
|  | Freeze | -0.12 | -0.44 | -0.14 | -0.29 | -0.20 | -0.37 |

989





**Table 3 Publication sources and abbreviations used in Fig. 3. Method of calculating melt rates is summarized as Ocean Model (OM), Standard Glaciological Method (GM), Ocean Observation (OO), Geophysical Tracer (GT), following the nomenclature used in Table S2 of Rignot et al. (2013) supplementary document. The time period of observation(s) or forcing files are listed together with source of data or model output.**

| Pub. Abbr. | method | reference | time period | estimate source |
|---|---|---|---|---|
| TS | OM | this study | NA | numerical model |
| M14 | GM | Moholdt et al. (2014) | 2003-2009 | |
| R13 | GM | Rignot et al. (2013) | 2003-2009 | ICESat |
| | | | 2007-2008 | ALOS PALSAR |
| | | | | InSAR |
| | | | 1979-2010 | RACMO2 |
| | | | | Operation IceBridge |
| | | | | BEDMAP |
| D13 | GM | Depoorter et al. (2013) | 2003-2009 | ICESat |
| | | | 1994-2002 | ERS-1 |
| | | | 2007-2009 | ERS-2 |
| | | | 2007-2009 | InSAR |
| | | | 1979-2010 | RACMO2 |
| T12 | OM | Timmermann et al. (2012) | 1958-2010 | FESOM model, NCEP winds (1958-2010) |
| M11 | OM | Makinson et al. (2011) | NA | |
| H04 | OM | Hellmer (2004) | 1978-1997 | NCEP 10-m winds |



| | | | | |
|---|---|---|---|---|
| | | | | 2-m air temperature |
| | | | | specific humidity |
| | | | | cloudiness |
| | | | | and net precipitation |
| JP03 | GM | Joughin and Padman (2003) | 1997 | RADARSAT InSAR |
| N03 | OO | Nicholls et al. (2003) | 1995-1999 | CTD |
| F01 | OO | Foldvik et al. (2001) | 1992-1993 | CTD & mooring |
| G99 | OM | Gerdes et al. (1999) | NA | |
| G94 | GT | Gammelsrød et al. (1994) | Feb. 1993 | CFC-11, CFC-12, O2, Si |
| J92 | GM | Jacobs et al. (1992) | | |
| JD91 | GM | Jenkins (1991) | 1985-1988 | Radar echo sounding |
| S90 | GT | Schlosser et al. (1990) | Jan-Mar. 1985 | $\partial_{18}O$, He |

994
995




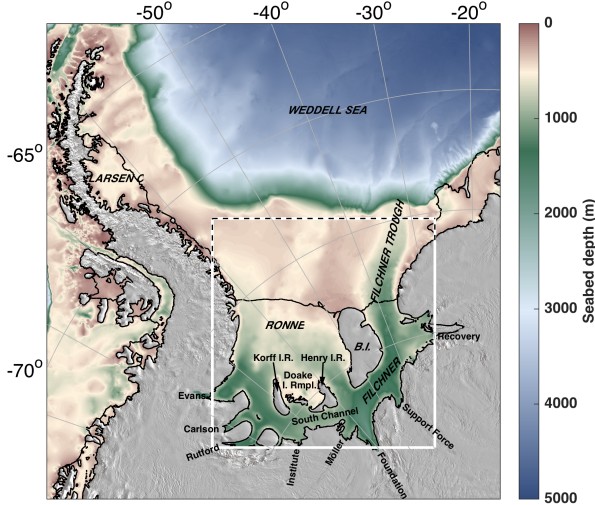

997

**Figure 1: Weddell Sea region of study with model domain outlined by white box. Dashed black lines highlight the open boundaries. The labels on land indicate the names of the tributary glaciers used for regional anslyses in this study. Black lines over seabed indicate the extend of th ice shelf and black lines around the grey mask indicate the ice sheet grounding line and/or transition between ocean and land.**

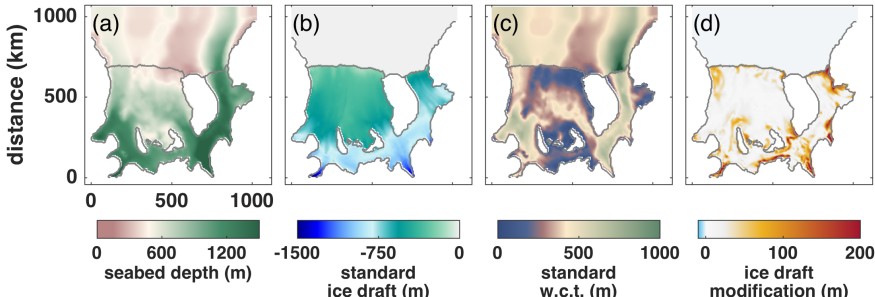

1002

**Figure 2: Bathymetry ($h$) and ice draft ($z_{ice}$) for the standard and modified geometries: (a) $h$ for both the standard and modified cases; (b) $z_{ice}$ for the standard case; (c) w.c.t. for the standard case; (d) Difference between standard $z_{ice}$ and modified $z_{ice}$, where difference > 0 indicates regions of melting and a corresponding decrease in $z_{ice}$ in the modified geometry when compared to the standard geometry.**





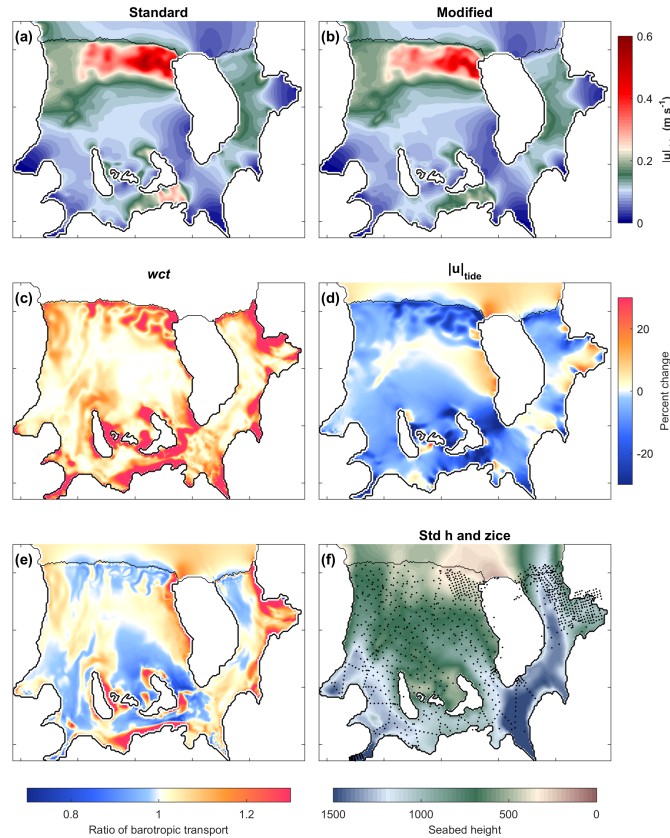

**Figure 3: (a) Barotropic current ($|\mathbf{u}|_{tide}$, Eqn. (5)) for the standard tides-only run. (b) Same as (a) but for the modified tides-only run. (c) Percent change in *wct* between the standard and modified cases with positive values indicating where the *wct* is greater in the modified geometry than the standard geometry. (d) Change in $|\mathbf{u}|_{tide}$ between (a) and (b) where Percent change > 0 indicates locations where the standard case $|\mathbf{u}|_{tide}$ is greater than the modified case $|\mathbf{u}|_{tide}$. (e) Ratio of barotropic transport ($wct \times |\mathbf{u}|_{tide}$) shown here as modified/standard, with values > 0 showing where there is increased transport for the modified case. (f) Seabed depth (as in Figure 2a) with existing seismic observation locations shown as black dots.**





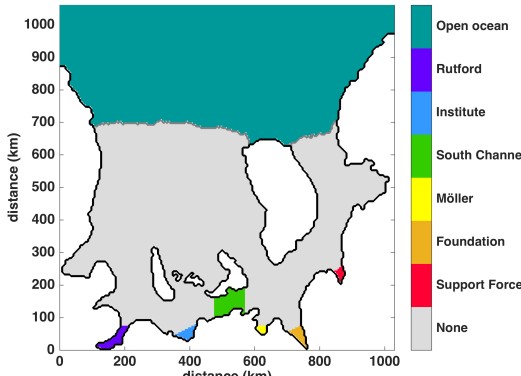


**Figure 4: Locations of the six meltwater dyes and the open ocean bulk dye explained in Sec. 2.3.3.**

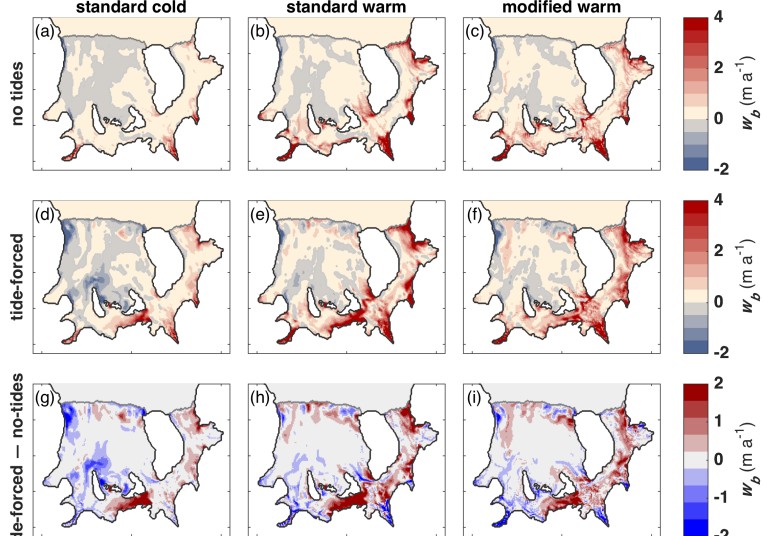


**Figure 5: Melt rates averaged over 1-year of steady state solutions for (top) no tides runs, (middle) tide-forced runs,**
**and (bottom) the difference between the tide-forced and no-tides melt solutions. Positive values in the bottom**
**subplots show where there is more basal melting in the solutions that include tidal forcing (middle subplots). The left**
**column (a, d, g) shows results for the standard cold case; the middle column (b, e, h) shows results for the standard**
**warm case; and the right column (c, f, i) shows results for the modified warm case.**



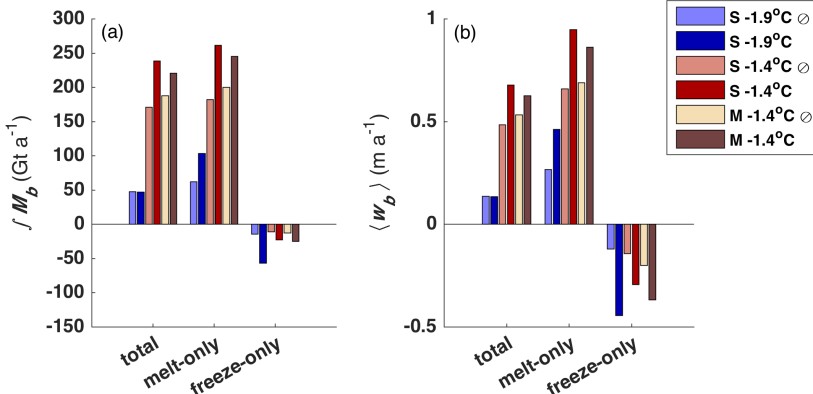


Figure 6: (a) integrated mass flux over "total" FRIS area, "melt-only" regions, and "freeze-only" regions for both
no-tide (⊘) and tide-forced cases. (b) Same regions and runs as in (a) but showing FRIS-averaged basal melting.

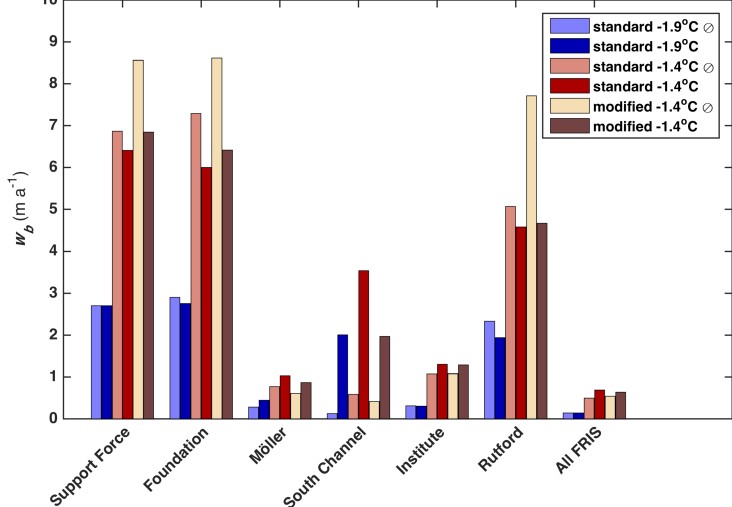


Figure 7: Melt rates averaged over last 12 months of steady state solutions in the standard cold, standard warm, and
modified warm cases for some of the regions shown in Fig. 4 and for both no-tides (⊘) and tide-forced simulations.
"All FRIS" duplicates the information shown by "total" in Fig. 6b.





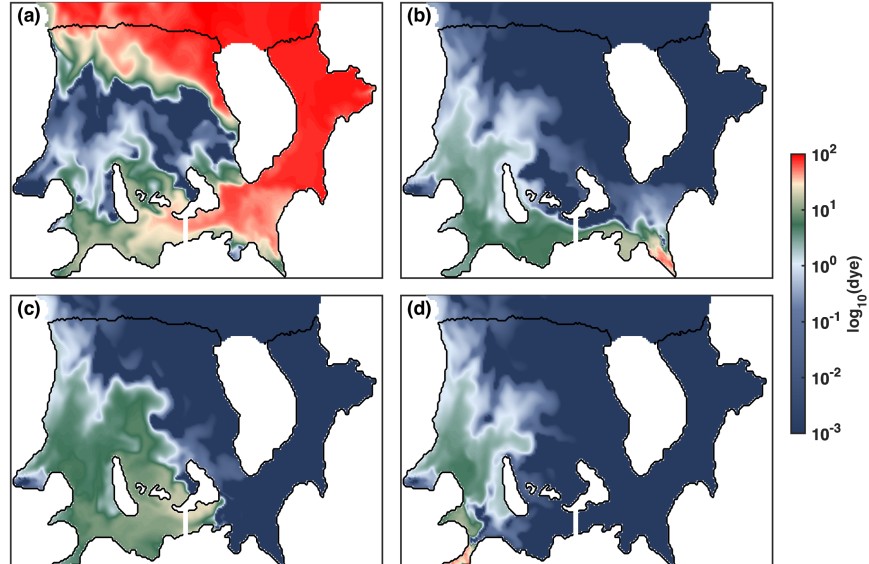


Figure 8: Distribution of dye concentration from the last time step and the upper model layer of the standard cold, tide-forced case described in Sect. 2.3.3. (a) bulk dye representing penetration of water initially north of the FRIS ice front. (b-d) meltwater dyes with sources in Foundation inlet, South Channel, and Rutford inlet, respectively (see Fig. 4 for dye release locations). The white line across South Channel represents the location of the transects in Fig. 13.



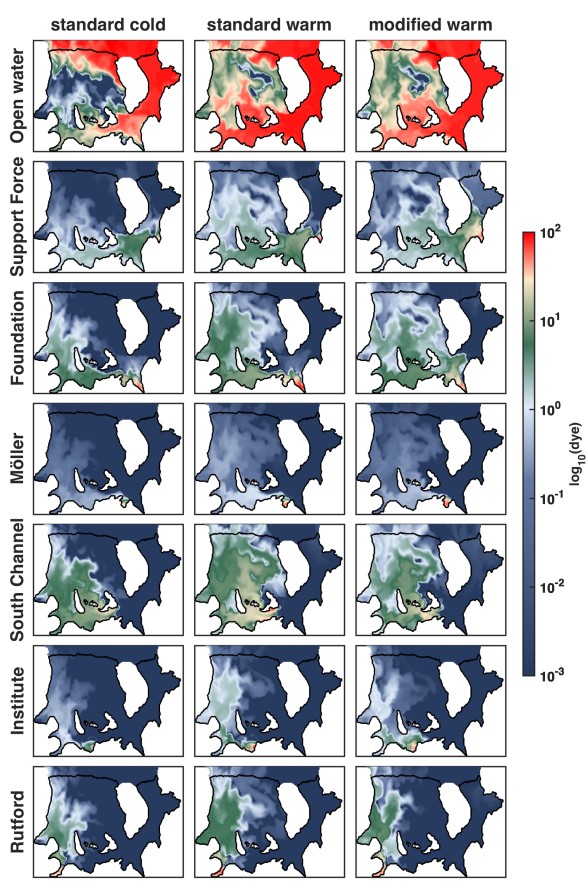

1036

Figure 9: An expanded distribution of dye concentration than that shown in Fig. 8 to include all meltwater dyes in
the three tide-forced base simulations. As in Fig. 8, dye concentrations are from the last 30-day average of upper
model layer solutions from the runs described in Sect. 2.3.3. The left hand column of this graphic includes the same
four regional plots as shown in Fig. 8.




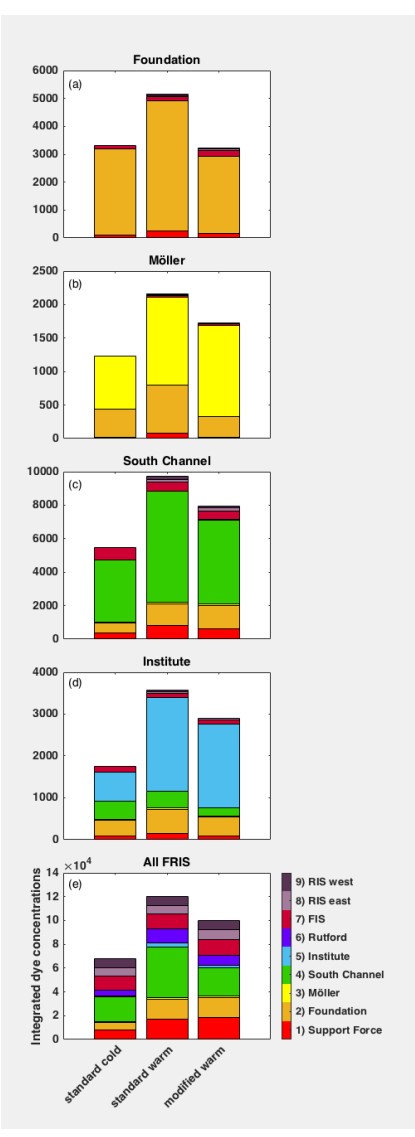


**Figure 10 Integrated meltwater dye (Sect. 2.3.3) by region for the tide-forced, base simulations, showing: (a) Foundation region, (b) Möller region, (c) South Channel region, (d) Institute region, and (e) all of FRIS. Regions are defined in Fig. 4.**






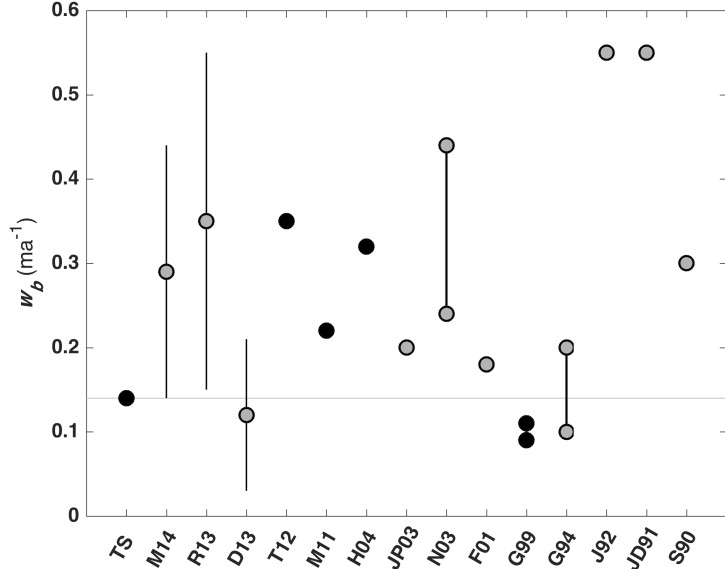


Figure 11 FRIS-averaged basal melt rate comparison between this study [TS] and others. Model results are shown
as black dots while observations are shown as grey dots. Error bars for remote sensing observations are shown as
black lines for M14 (Moholdt et al., 2014), R13 (Rignot et al., 2013) and D13 (Depoorter et al., 2013). Min and max
values are connected by thick, solid, black lines to show the range of values reported by N03 (Nicholls et al., 2003)
and G94 (Gammelsrød et al., 1994). A summary of the studies presented here and their abbreviations is provided in
Table 2.



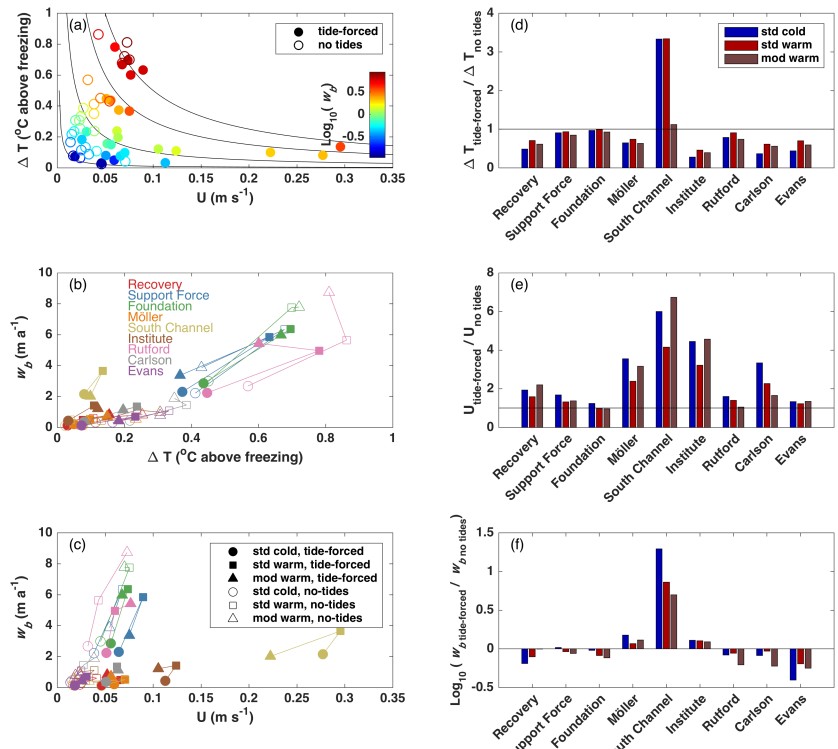

**Figure 12: Regional influences of $\Delta T$ and current speed on melt rates.** Tide-forced cases are plotted using solid marker style, e.g. "■", and no-tide cases are plotted using open marker style, e.g. "□". **(a)** Current speed ($U$, Eqn. (6)) vs. thermal forcing ($\Delta T$), color-coded according to melt rate ($w_b$). Black contours follow $\Delta T = c/U$ (with c being a set of different scalars), along which costant values of $w_b$ (as in Sect. 2.3.1) are expected to be found. **(b)** $\Delta T$ vs. $w_b$ for each region. **(c)** $U$ vs. $w_b$ for each region. **(d)** $\Delta T$ difference between no-tides and tide-forced cases such that positive values show where thermal forcing is stronger in the no-tides cases, **(e)** current speed difference between tide-forced ($U_{tides}$) and no-tides ($U_{no\ tides}$) cases, and **(f)** $w_b$ difference between tide-forced and no-tides cases.



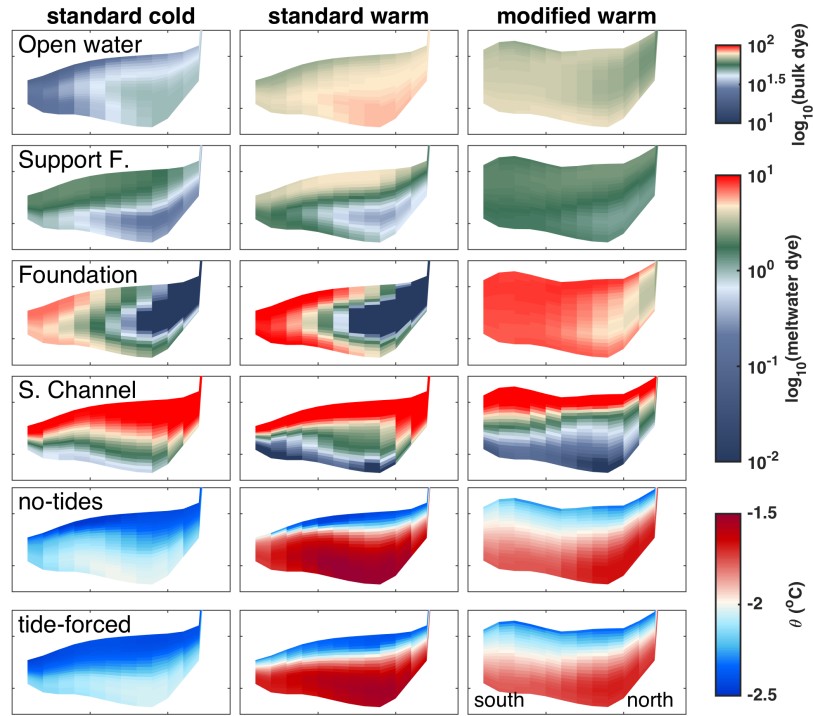

1061

Figure 13: Transects of dyes (upper four rows) and potential temperature ($\theta$, lower two panels) across South
Channel at the western tip of Henry Ice Rise. The upper four panels are for tide-forced runs only. Transect location
is shown in Fig. 8.