# Peer review of "Tidal influences on a future evolution of the Filchner-Ronne Ice Shelf cavity in the Weddell Sea, Antarctica"

_The Cryosphere, 2017_

## Referee Comment (RC1) · HH Hellmer (Referee) · 23 Aug 2017

General comments:

The authors present a very comprehensive study of the influence of tides (K1, O1, M2, S2) on the circulation beneath, melting at the base, and spreading of glacial melt within the cavity of the largest - by volume - Antarctic ice shelf. This has been done, switching the tidal forcing on/off, for two different initial ocean temperatures, today and moderate warming of the cavity, and a modified cavity geometry, which considers the reduction of ice shelf thickness due to increased melting. Despite the large number of simulations the reader does not get lost, since the presentation of the results is well

organized, allowing for an easy 'digestion' of the findings. The study is very timely not only in view of an increasing awareness that the ocean is the prime driver of the Antarctic Ice Sheet mass loss but also in view of the long-existing assumption, based on a few model exercises (e.g., MacAyeal 1985), that tides might play a significant role in the basal processes, though not being considered in present state-of-the-art coupled ocean-ice shelf-models. This paper demonstrates that the incorporation of tides is necessary to correctly simulate the melt/freez pattern in a large 'cold cavity' because of the consequences it has for the dynamics of the ice shelf and thus the whole ice sheet, not only for today but also in view of the changes ice shelf cavities might face in a warming climate.

The results, however, should be viewed more in a qualitative manner, due to several model shortcomings with regard to the set up and the forcing, but here the model is in good company with other 'more sophisticated' models. I especially like the study of the interplay between different ice stream inlets and their dependency on location and strength of upstream melting.

In summary, I strongly recommend publication in TC after the comments have been considered and corrections made.

Specific comments:

L032 – How can a paper published in 2006 cover the period 2002-2016? Actually, according to Monaghan et al. (2006), which covers the period 1955-2004, small changes in SA only occurred on the EAIS.

L055 – With regard to the direct flow of CDW into the ice shelf cavities of ABS Jenkins et al. (2010) is a more appropriate reference.

L081 – More precise: Hellmer et al. (2017) not just reversed the atmospheric conditions to a colder state but to 20th -century conditions.

L164 - Equation (4) must read wb = QT0/L x rho . And, somewhere it should be

mentioned that heat flux through the ice shelf is ignored.

L216 – It comes as a surprise that open ocean wct = h has a maximum value of 1914 m, though the model domain (Fig. 1) only covers the southern Weddell Sea continental shelf.

L283 – It is not clear whether the 'bulk dye' was added to the whole water column or just to the suface-sigma layer. The confusion starts when looking at Fig. 8a, which looks more like a 'bottom dye' distribution.

L305 – It is not clear what is meant with 'outer/interior grounding line' – please explain.

L335 – Here, a serious deficit of the model becomes obvious, since the refreezing along the eastern coast of Berkner Island (e.g., Rignot et al. (2013)) is missing, certainly because the model exaggerates the flow into the Filchner cavity.

L396 – A reduction of area-averaged melt rates due to adding tides also happens for Support Force.

L426 – The comparison with Hellmer et al. (2012) is risky because this kind of circulation only happens for the 'warm phase', while here the same circulation pattern exists for the 'standard cold case'. Such comparison might provoke a critical reader to question ROMS' performance in general.

L714 – By summarizing important results I miss (5): The increase of refreezing in central RIS in the 'standard cold case', representing today's conditions. This is an important finding because refreezing in this area certainly changes the dynamics of the ice shelf by increasing the buttressing around Henry and Korff.

L1044 – Please explain the difference between 'region' and 'inlet', used in Fig. 8.

L1061 – Please explain why Foundation shows a high dye concentration at the bottom. The signal cannot be advected from Support Force because there it does not exist, and highest melting beneath Foundation should stabilize the water column such that most,

if not all, of the dye should concentrate at the base.

Technical corrections:

L031 – The dominant terms in the Antarctic ICE SHEET (AIS) mass budget....

L060 – We focus here ON . . .

L267 - . . . over the LAST 30 days?

L298 - . . . ice shelf frontal zone (ISFZ) of the RIS (here, front is redundant)

L337 - ...northWESTERN RIS.

L466 - . . . from all upstream sourceS.

L510 – Fig. 10e

L 833 – HelLmer, H. H.

L1000 - .. the extent of thE ice shelf...

L1017 – Locations of the six meltwater dye RELEASES..

References: Jenkins, A. et al. (2010), Nature Geoscience, doi:10.1038/ngeo890

MacAyeal, D. R. (1985) The evolution of tidally triggered meltwater plumes below ice shelves. Oceanology of the Antarctic Continental Shelf, ed. S. S. Jacobs, Antarctic Research Series, 43, 133 – 143, American Geophysical Union, Washington D.C.

---

## Referee Comment (RC2) · K. Makinson (Referee) · 7 Sep 2017

Review Journal: TC Title: Tidal influences on a future evolution of the Filchner-Ronne Ice Shelf cavity in the Weddell Sea, Antarctica Author(s): Rachael D. Mueller et al. MS No.: tc-2017-110

The authors present a modelling study of how tides influence the basal melting and freezing beneath Filchner Ronne Ice Shelf, the circulation of melt water within the cavity, and subsequent evolution of the cavity shape. Multiple simulations are presented with and without tidal forcing, at two different ocean temperatures, and with a modified cavity geometry that is generated from enhanced melting in the warmer scenario. The

paper has a logical structure and the good use of tables and multi pane figures assists the reader through the various simulations. The results clearly highlight that modelling ocean tides in this region is critically important as they impact on the magnitude of basal melting (and freezing) which in turn influences the buoyancy driven circulation with the cavity. This result supports the general conclusions of other modelling studies looking at FRIS, but by considering warmer ocean scenarios and a melt/freeze modified cavity geometry, this study provides insight into how tidal currents and circulation patterns will change and feed back into modifying the ice shelf and ice sheet. While present day geometries and ocean temperatures are used for forcing, the simple uniform ocean salinity north of FRIS ignores the present day east to west increase in density. As a consequence, the overall modelled circulation within the cavity is reversed from the present day conditions. The circulation scenarios presented here are therefore more representative of the Hellmer et al warm phase and having the label of 'present day' is misleading when talking about the circulation. It would be useful to clarify this point early in the paper for readers who are less familiar with this region.

Overall this is a useful contribution to the suite of modelling efforts focused on this region and recommend publication in TC.

Suggested minor revisions:

L39 remove 'the'

L58 change to 'over the observational record.'

L60 change 'the' to 'on'

L69 remove both 'the'

L71 Modified Warm Deep Water (MWDW)?

L73 MWDW also seen at Ronne Ice Front (Mooring R2 and CTD's Foldvik et al 2001 doi:10.1029/2000JC000217)

L90 change to 'pattern and magnitude of'

L118 Are these temperatures and salinities restored throughout the model runs?

L122 It would be worth mentioning the lack of an east to west density gradient and hence the reverse circulation in the cavity.

L195-196 it would be useful to define what these numbers (Haney number and Beckmann and Haidvogel number), are if you are going to mention them.

L203 extreme stratification profile – a little more explanation maybe?

L212 Mention that this is a known problem/limitation with this type of model.

L216 1914 m. correct?

L233 .03 is this correct? Remove 'below'

L234 remove 'above freezing'

L236 consider removing ,. Given….bathymetry.'

L239-40 consider changing to 'conditions and ran with and without . . .'

L243 change to 'tide-resolving with at temporal resolution of 2 hours and'

L270 change 'over' to 'after'

L286 section label needed

L298 change to 'maximum tidal currents along'

Para L327-330 consider adding a value or percentage for the 'increased melt rates' and 'slightly modifies'

L336 it should be noted that there is very limited freezing north of Henry ice rise. L415 'southern' rather than 'innermost'

L441 'these' to 'the'

L466 sources

L510 Fig. 10e

L543 'is' to 'are'

L706 Ronne Ice Front

L759 Any bathymetry data from beneath ice shelves is useful as it will help better define the cavity geometry which you have demonstrated to be important for the whole system (tides and circulation).

Fig 1. Add W and S labels for lon and lat.

L1010 remove 'than the standard geometry'

L1012 change to 'barotropic tidal transport'

L1017 change to 'continuous dye release'

Fig8 and 9. Mention in caption that this is after 2 years.

Fig12. Holland et al 2008 show similar melt figures over an extended range. doi:10.1175/2007jcli1909.1

Also Holland et al 2007 doi:10.1029/2006JC003915 show similar results to your dye tracer experiments.

---

## Short Comment (SC1) · 28 Sep 2017

Dear Dr. Mueller and coauthors,

I am not a reviewer of this paper, but I would like to pick the rarely used opportunity for a discussion including more participants than just authors and reviewers.

I think this is a very interesting study, rightfully questioning one of the conclusions drawn from the coupled ice sheet – ocean model RAnGO (Timmermann & Goeller, Ocean Science 2017). TG17 state that in a warm-water-inflow scenario with thinning ice shelf, the increasing ice base slope and increasing water column thickness tend

to increase melt rates due to an easier supply of warm(ing) water to the (still) deepest parts of the ice shelf. Now, this study here reminds us that tidal velocities may decrease as an answer to increasing wct, which may partly, fully or over-compensate the effect that we found to be dominant in RAnGO. I find myself wondering how much trouble it would be to run the tidal model for ice shelf geometries varying on a year-to-year basis, let's say as part of a coupling interface between ice shelf and ocean, and then use the tidal velocities as an additional forcing field for the ocean model. I am not arguing this should be discussed in the paper; it's just a tought that crossed my mind.

Congratulations to this nice piece of work!

Respectfully yours, Ralph Timmermann

---

## Author Comment (AC1) · 28 Oct 2017

We are grateful to Drs. Keith Makinson and Hartmut Hellmer for their thorough and thoughtful reviews.  This manuscript will be much improved by their input.  We have made changes to our document and are including below an overview of these changes.

Referee comments are in **bold**, responses are in *italics*, and corrected grammar indicated by .

**Hartmut Hellmer**

**Specific comments:**

**L032 – How can a paper published in 2006 cover the period 2002-2016? Actually, according to Monaghan et al. (2006), which covers the period 1955-2004, small changes in SA only occurred on the EAIS.** *Thank you for pointing out this oversight. We have rewritten this text and provided an updated ref. (Wang et al., 2016).*

**L055 – With regard to the direct flow of CDW into the ice shelf cavities of ABS Jenkins et al. (2010) is a more appropriate reference.** *Corrected (L67). We have also added a cite to Jacobs et al. (2013), since Jenkins et al. (2010) only discusses measurements for Pine Island Glacier ice shelf, while Jacobs et al. (2013) demonstrates CDW flows into Getz (also ABS) as well.*

**L081 – More precise: Hellmer et al. (2017) not just reversed the atmospheric conditions to a colder state but to 20th -century conditions.** *We have augmented this section to include this clarification.*

**L164 -  . And, somewhere it should be mentioned that heat flux through the ice shelf is ignored.** *Thank you for catching this typo!  We have added a statement regarding the heat flux through the ice to the paragraph preceding Eqn. (1), and stated that we have ignored it for this study (reasonable for thick ice).*

**L216 – It comes as a surprise that open ocean wct = h has a maximum value of 1914m, though the model domain (Fig. 1) only covers the southern Weddell Sea continental shelf.** *Thanks.  The value is now corrected, to 1211 m in Filchner Trough.*

**L283 – It is not clear whether the 'bulk dye' was added to the whole water column or just to the surface-sigma layer. The confusion starts when looking at Fig. 8a, which looks more like a 'bottom dye' distribution.** *The dye is added uniformly to all levels.  We have amended the text to state "The bulk dye was initialized at a concentration of 100% over the entire water column but was not replenished after these simulations began.". We have included a couple of figures here to demonstrate the vertical structure of the bulk dye initialization (*Figure 1*) as well as the dye distribution in the surface and bottom layers.*

[Figure]

**Figure 1 Dye concentration from 0-100% in sigma levels 1 (bottom), 5, 15, and 24 (surface)**

[Figure]

(a)  (b)

**Figure 2 Surface and bottom level dye distribution for the standard cold case, with (b) corresponding to Fig 8a and the upper left panel of Fig. 9.**

**L305 – It is not clear what is meant with 'outer/interior grounding line' – please explain.** *The labeling has been corrected using geographical referencing ("eastern", "southeastern", etc). We hope this helps to clarify the intended meaning.*

**L335 – Here, a serious deficit of the model becomes obvious, since the refreezing along the eastern coast of Berkner Island (e.g., Rignot et al. (2013)) is missing, certainly because the model exaggerates the flow into the Filchner cavity.** *We have clarified in the text how our lack of density gradient along the continental shelf yields a circulation bias with clockwise flow around Berkner Is., rather than the observed counterclockwise flow that is represented by models that include horizontal density gradients across the front.*

**L396 – A reduction of area-averaged melt rates due to adding tides also happens for Support Force.** *Support Force is now included in this statement.*

**L426 – The comparison with Hellmer et al. (2012) is risky because this kind of circulation only happens for the 'warm phase', while here the same circulation pattern exists for the 'standard cold case'. Such comparison might provoke a critical reader to question ROMS' performance in general.** *The intention of this phrasing was to highlight that in both these results and in the Hellmer et al. (2012) results the circulation within the FRIS cavity is clockwise, such that the primary ocean inflow is through FIS; but(!),we have decided to take a different approach in describing this circulation. The Discussion section now includes an overview of how our standard cold results are to be interpreted in the context of known present-day circulation. This addition also aims to address a comment by Keith Makinson.*

> Our results show that tide forcing is important to FRIS ice-ocean interactions over a range of initial temperatures and with large variations in regional impacts. The aim of these simulations was to apply, uniformly, temperatures and salinities that approximate (1) present-day, inflow conditions and (2) a representative temperature of future, inflow conditions, reflecting a modest increase in temperature; however, the circulation within this particular ice shelf cavity is strongly affected by sea-ice formation on top of a general circulation that establish an east to west density gradient across the continental shelf (e.g. Foldvik et al., 1985; Nicholls et al., 2009). As such, our "present-day" scenario is a hypothetical one that forms the basis of this sensitivity study but that should not be interpreted to reflect the known circulation within the cavity. The known circulation within the cavity, setup by the east to west density gradient on the continental shelf, has inflow in the Ronne Depression and a counter clockwise circulation around Berkner Is. (Foldvik et al. 2001). Our simulations for both the present-day and melt-adjusted cases predict inflow through the FIS and a clockwise circulation around Berkner Is. This pattern of circulation reflects the future warming scenario presented in Hellmer et al. (2012). This sensitivity study, therefore, does not include changes that would occur from a shift in cavity circulation from a scenario that has the east to west density gradient along the continental shelf to one where that density gradient is relaxed to the degree that the sub-ice-shelf cavity circulation would change. Although these simplifications restrict the predictive capacity of this study, they do not much affect the results of our sensitivity analysis on ice-ocean interactions within FRIS and in particular the feedbacks found between tides and changing geometry and implications for further research.

**L714 – By summarizing important results I miss (5): The increase of refreezing in central RIS in the 'standard cold case', representing today's conditions. This is an important finding because refreezing in this area certainly changes the dynamics of the ice shelf by increasing the buttressing around Henry and Korff.** *This result has already been reported in Makinson et al. (2011) for present-day ocean conditions; we therefore added the contribution of our work, with appropriate attribution to Keith Makinson for his, We clarified our results in the context of previous work by introducing a new Discussion section that focuses on basal freezing beneath FRIS (**Sect. 4.5**)*

**4.5 Implications of regional freeze conditions on ice sheet mass balance**

As described in **Sect. 3.2.1** and **Sect. 3.2.2**, refreezing occurs in our simulations throughout a large region of the central RIS. Refreezing in this region is qualitatively consistent with estimates of basal mass balance from satellite-based remote sensing (e.g., Joughin and Padman, 2003; Rignot et al., 2013; Moholdt et al., 2014). Persistent refreezing along ice flowlines can create a marine ice layer up to hundreds of meters thick, as observed in ice cores (Engelhardt and Determann, 1987; Oerter et al, 1992), and in radio-echo sounding and seismic measurements (Joughin and Vaughan, 2004; Lambrecht et al., 2007). These observations are important in the context of other studies which show that marine ice accretion supports ice shelf stability (Kulessa et al., 2014; McGrath et al., 2014; Li et al. (submitted)). Our standard cold tide-forced case produces local maxima in marine ice growth rates in the northwestern RIS, the region northeast and east of Korff Ice Rise, and the region to the north and west of Henry Ice Rise (**Fig. 5d**). The spatial pattern of these freeze conditions differs from observed patterns (Joughin and Padman, 2003; Rignot et al., 2013; Moholdt et al., 2014). We attribute this difference to the consequences of omitting the east to west density gradient along the continental shelf.

 The regions of freezing are broadly consistent in all our model runs (**Fig. 5**) and the net mass increase in refreezing regions are increased when tide forcing is added (**Fig. 6**). Our standard cold tide-forced case has a ~4-fold mass gain compared to the standard cold no-tides case; this result is consistent with Makinson et al. (2011). In both warm cases, standard and modified geometry, adding tides increases net marine ice formation by a factor of two. That is, tides will continue to be important for marine ice accretion beneath FRIS if ocean temperatures rise as predicted by Hellmer et al. (2012, 2017) and will, therefore, continue to play a role in FRIS ice shelf stability.

*We also included the following statement as (3) in our list of conclusions:*

(3) Adding tide forcing increases the overall freezing conditions for all three cases including the two warm cases. Since freeze conditions lead to marine ice accretion, and marine ice strengthens the ice shelf, the tidal contribution to ice-shelf dynamics is expected to continue through future ocean warming, increasing grounding and associated contact stresses in the region near the Henry and Korff ice rises and Doake Ice Rumples

**L1044 – Please explain the difference between 'region' and 'inlet', used in Fig. 8.** *We have corrected the use of "inlet" as shorthand for "inlet region" by changing "inlet" to "region", while also changing the parenthetical statement "(see Fig. 4 for dye release regions)" in order to make this phrasing consistent.*

**L1061 – Please explain why Foundation shows a high dye concentration at the bottom. The signal cannot be advected from Support Force because there it does not exist, and highest**

**melting beneath Foundation should stabilize the water column such that most, if not all, of the dye should concentrate at the base.** *We puzzled over this for a while. Based on the following figure (Fig. 3, below), and other analyses of T(z), S(z), and FIS dye profiles, we conclude that FIS dye is present at depth in the S. Channel due to the combined influence of mixing within the Möller inlet region as well as a shoaling of bathymetry in both Möller and South Channel.  FIS dye appears to circulate first into the Möller inlet region where additional freshwater is added to the "upper" branch of FIS-doped water and where mixing processes transport dye to depths that correspond to bottom level depths within the South Channel. Within South Channel, all water contains meltwater from different sources, including locally. The density stratification depends on the sum of freshwater from all sources, whereas a specific dye (e.g., FIS), depends only on meltwater from that region.*

*In order to clarify this influence, we have augmented section 4.3, to include the following statement:*

> The presence of Foundation dye in the bottom level of the S. Channel transect reflects a shoaling of bathymetry (**Fig. 2c**) and mixing with the Möller region that allows the dye to be distributed to the bottom level within the Möller inlet region and then advected, at depth, through South Channel.

The evidence for this statement can be seen in Figure 3, below.  Foundation dye is shown in the bottom, mid-depth and surface levels of the standard cold case. We attribute the circulation at this depth to be driven by changes in bathymetry, as shown in Fig. 2 of the main manuscript and here by w.c.t.

[Figure]

Figure 3: Maps of Foundation dye taken at the bottom level (N=1), a mid-depth level (N=15) and the surface level (N = 24) for the standard cold cases with water column thickness on right (as a duplicate of Fig. 2c, in main manuscript).

**Technical corrections:**

**L510 – Fig. 10e**

**L 833 – HelLmer, H. H.**

**L1000 - .. the extent of thE ice shelf...**

**L1017 – Locations of the six meltwater dye RELEASES.** *We have included this change together with an emphasis on "model dye", which isn't really "released" per se, though we agree with this choice in verb.*

---

## Author Comment (AC2) · 28 Oct 2017

We are grateful to Drs. Keith Makinson and Hartmut Hellmer for their thorough and thoughtful reviews. This manuscript will be much improved by their input. We have made changes to our document and are including below an overview of these changes.

Referee comments are in **bold**, responses are in *italics*, and corrected grammar indicated by

**Keith Makinson**

**Suggested minor revisions:**

**L118 Are these temperatures and salinities restored throughout the model runs?** *Yes. We have augmented the text to explain:* "Model hydrography is restored to these initial temperatures and salinity along the boundaries using a mixed radiation and nudging condition (Marchesiello *et al.*, 2001) over a 20-day period."

**L122 It would be worth mentioning the lack of an east to west density gradient and hence the reverse circulation in the cavity.** *We agree with the need to highlight this limitation in our model setup but feel that this information is best contained in the discussion section rather than the methodology section. We now use this information to introduce the discussion section and establish how our results can be interpreted in the context of other studies. The paragraph in question in our methodology section now reads:*

> Our simulations were initialized with a homogeneous, stationary ocean that has a potential temperature of either $\theta_{init}$ = -1.9°C ("cold case") or $\theta_{init}$ = -1.4°C ("warm case"). Initial salinity is defined as $S_{init}$ = 34.65 for all cases. Model hydrography is restored to these initial temperatures and salinity at the boundaries using a mixed radiation and nudging condition (Marchesiello *et al.*, 2001) over a 20-day period. The standard geometry cold case incorporates a uniform temperature and salinity that approximates conditions of the primary water mass entering the ice shelf cavity (Foldvik et al., 2001; Nicholls et al., 2001, 2009). The consequences to FRIS cavity circulation in choosing a uniform $\theta_{init}$ and $S_{init}$ are discussed in **Sect. 4**. The warm case represents a moderate ocean warming scenario with an increase of 0.5°C in the temperature of water being advected into the FRIS cavity. This change is much smaller than the 2°C temperature increase in the inflowing water by the end of this century predicted by Hellmer et al. (2012), but was chosen to investigate whether initial

feedbacks due to melt-induced changes in cavity shape from initial warming might be positive or negative. Our idealized simulations do not include wind forcing, frazil ice, or sea-ice formation.

*The intro to the discussion section (Sect. 4) goes into more details as follows.*

Our results show that tide forcing is important to FRIS ice-ocean interactions over a range of initial temperatures and with large variations in regional impacts. The aim of these simulations was to apply temperatures and salinities that approximate (1) present-day inflow conditions and (2) a representative temperature of future inflow conditions, reflecting a modest increase in temperature. The choice of spatially constant initial temperature ($\theta_{\text{init}}$) and salinity ($S_{\text{init}}$) does, however, influence circulation into and under the FRIS cavity. In the real ocean, spatial structure of the wind stress and production of dense HSSW by sea ice formation over Ronne Depression establishes an east-west density gradient across the continental shelf (e.g. Foldvik et al., 1985; Nicholls et al., 2009) that leads to stronger flows into the cavity across the RIS front than our model generates. Our "present-day" scenario should, therefore, be regarded as the basis of this sensitivity study rather than a prediction of known circulation within the cavity. In particular, our standard cold case misses the inflow in the Ronne Depression and a counter-clockwise circulation around Berkner Island (Foldvik et al. 2001). In contrast, our simulations for both the present-day and melt-adjusted cases predict the primary inflow through the FIS and a clockwise circulation around Berkner Island; this pattern of circulation is, however, consistent with the future warming scenario presented in Hellmer et al. (2012). The fundamental conclusions of our sensitivity analysis of ice-ocean interactions within FRIS are, however, independent of these differences from the real-world modern circulation.

**L195-196 it would be useful to define what these numbers (Haney number and Beckmann and Haidvogel number), are if you are going to mention them.**

*The paragraph in question now includes the following description:*

The ice draft and bathymetry were each smoothed to minimize errors in the baroclinic pressure gradient that arises with the terrain-following coordinate system used in ROMS (Beckmann and Haidvogel, 1993; Haney, 1991). The two parameters used to quantify smoothing are the Beckmann and Haidvogel number, rx0 = |h(e) – h(e')| / (h(e) + h(e') (Beckmann and Haidvogel, 1993), and the Haney number, rx1 = |h(e, k) – h(e', k) + h(e, k-1) – h(e', k-1)| / (h(e, k) + h(e', k) – h(e, k-1) – h(e', k-1)) (Haney, 1991), where $1 \leq k \leq N$ and e and e' represent two adjacent cells. Together, these parameters establish that the surface (ice) and bottom bathymetry slopes are sufficiently small to reduce or eliminate spurious flows due to a horizontal pressure gradient and ensures hydrostatic consistency throughout the water column at adjacent horizontal grid nodes. Our Beckman and Haidvogel number, rx0, is less than 0.045 along both surface and bottom topographies, and our Haney number, rx1, is less than 10 in both surface and bottom levels except for

some areas along the ice shelf front, where rx1 is larger and reaches a maximum value of 17.

**L203 extreme stratification profile – a little more explanation maybe?** *The sentence now reads: "We initialized these models with horizontally uniform temperature and salinity fields taken from a standard warm case profile, in the vicinity north of Bjerkner Island, where the strongest stratification of all runs is represented."*

**L212 Mention that this is a known problem/limitation with this type of model.** *We have included this information.*

**L233 .03 is this correct? Remove 'below'.** *Yes.  0.03 is correct for the mode, but we agree that this number is a questionable asset to the discussion.  We have changed the text to include information on the median: "...and a median of 14 m". A graphic of the distribution is included below.*

[Figure]

**L286 section label needed.** *Indeed! We have corrected this and other Section label errors.*

**Para L327-330 consider adding a value or percentage for the 'increased melt rates'**

**and 'slightly modifies'.** *We have modified the text to read:*

> The pattern of $w_b$ for the standard warm no-tides case (**Fig. 5b**) is generally similar to the standard cold no-tides case (**Fig. 5a**) but with a 3.5 fold increase in the shelf-averaged value. Changing cavity shape while imposing the same initial ocean temperature in the modified warm case (**Fig. 5c**) only slightly reduces melt rates (by 10%) for the no-tides scenario (cf. **Fig. 5b and 5c**).

**Fig12. Holland et al 2008 show similar melt figures over an extended range. doi:10.1175/2007jcli1909.1** *This is an important paper to cite here, and we have done so as follows: "*In general, our values are in range of those shown in Holland et al. (2008), (c.f. their Fig. 1 and our Fig. 12b),…."

**Also Holland et al 2007 doi:10.1029/2006JC003915 show similar results to your dye tracer experiments.** *We have included a reference to this important paper in the first paragraph of section 2.11: "*Our model excludes the influences of wind-driven circulation and sea-ice formation. As such, it is perhaps no surprise that these dye distributions are qualitatively similar to those shown by Holland et al. (2007)."

---

## Author Comment (AC3) · 29 Oct 2017

Dear Dr. Timmermann:

We are grateful for your comments and interest in ongoing discussions regarding the advancement of model predictions inclusive of the influence of tides.

As you highlighted in your comment, the results of our study together with your Timmermann & Goeller (2017) results demonstrate a need to continue advancing our understanding of how warmer water will affect change within the FRIS cavity.

With regard to updating tides as a model progresses: The easiest way is to always use

explicit tidal forcing at the open ocean boundaries, and u*-dependent ocean/ice-shelf thermodynamics. Then, at least until the cavity tides start to also change the global tides (which happens), you always have real tidal forcing and the cavity tidal currents will change as needed.

This is a great topic for ongoing discussion and research. We are keen to work with you and others toward resolving these uncertainties in future work!

Thank you for your comment.

Sincerely, Rachael Mueller (on behalf of all authors)
* * *